# Relief of autoinhibition by conformational switch explains enzyme activation by a catalytically dead paralog

Oleg A Volkov[1], Lisa Kinch[2], Carson Ariagno[1], Xiaoyi Deng[1], Shihua Zhong[1], Nick Grishin[2,3], Diana R Tomchick[2], Zhe Chen[2], Margaret A Phillips[1,4]*

[1]Department of Pharmacology, University of Texas Southwestern Medical Center, Dallas, United States; [2]Department of Biophysics, University of Texas Southwestern Medical Center, Dallas, United States; [3]Howard Hughes Medical Institute,University of Texas Southwestern Medical Center, Dallas, United States; [4]Department of Biochemistry, University of Texas Southwestern Medical Center, Dallas, United States

**Abstract** Catalytically inactive enzyme paralogs occur in many genomes. Some regulate their active counterparts but the structural principles of this regulation remain largely unknown. We report X-ray structures of *Trypanosoma brucei S*-adenosylmethionine decarboxylase alone and in functional complex with its catalytically dead paralogous partner, prozyme. We show monomeric *Tb*AdoMetDC is inactive because of autoinhibition by its N-terminal sequence. Heterodimerization with prozyme displaces this sequence from the active site through a complex mechanism involving a *cis*-to-*trans* proline isomerization, reorganization of a β-sheet, and insertion of the N-terminal α-helix into the heterodimer interface, leading to enzyme activation. We propose that the evolution of this intricate regulatory mechanism was facilitated by the acquisition of the dimerization domain, a single step that can in principle account for the divergence of regulatory schemes in the AdoMetDC enzyme family. These studies elucidate an allosteric mechanism in an enzyme and a plausible scheme by which such complex cooperativity evolved.

*For correspondence: margaret.
phillips@utsouthwestern.edu

**Competing interests:** The authors declare that no competing interests exist.

## Introduction

The availability of numerous sequenced eukaryotic genomes has uncovered enzyme paralogs across diverse gene families that are predicted to be enzymatically inactive because they lack essential catalytic residues (*Adrain and Freeman, 2012*; *Pils and Schultz, 2004*; *Todd et al., 2002*; *Reynolds and Fischer, 2015*; *Reiterer et al., 2014*; *Kung and Jura, 2016*). These 'pseudoenzymes' are estimated to represent up to 10% of human encoded proteins, and are particularly abundant within the protease and kinase families. The roles of pseudoenzymes have been delineated in only specific cases, but the general principles by which they might contribute to organismal fitness remain incomplete. An interesting hypothesis emerges from the fact that many enzymes form functional oligomers (*Goodsell and Olson, 2000*; *Marianayagam et al., 2004*). This property leads to the idea that pseudoenzymes might generally evolve to serve as regulators of enzymes, directly interacting with their cognate active homolog to exert regulatory control.

A case study of pseudoenzyme regulation is found in the parasitic trypanosomatids, eukaryotic pathogens that cause human African trypanosomiasis (HAT), Chagas disease, and Leishmaniasis, globally infecting up to 20 million people (*Stuart et al., 2008*). Trypanosomatids encode inactive paralogs for two essential enzymes in the polyamine pathway (*Willert et al., 2007*; *Willert and Phillips, 2008, 2012*; *Nguyen et al., 2013*). These enzymes, *S*-adenosylmethionine decarboxylase

(AdoMetDC) and deoxyhypusine synthase, have been characterized from *Trypanosoma brucei*, the causative agent of HAT. Both enzymes are profoundly activated (~1000 fold increase in catalytic efficiency) by oligomerization with their paralogous pseudoenzyme leading to formation of catalytically functional complexes.

AdoMetDC is a pyruvoyl-dependent enzyme that catalyzes formation of decarboxylated AdoMet, a substrate required for biosynthesis of the polyamine spermidine from putrescine (*Pegg, 2009b*) (*Figure 1*). Spermidine is essential in all eukaryotes for hypusine modification of the translation elongation factor eIF5A by deoxyhypusine synthase (*Dever et al., 2014*). As a consequence the polyamine biosynthetic pathway has been targeted for development of anti-proliferative agents, including for the treatment of HAT (*Jacobs et al., 2011*; *Willert and Phillips, 2012*). The AdoMetDC pyruvoyl group plays a key role in the catalytic mechanism and derives from an autocatalytic processing reaction that cleaves the peptide backbone into β- and α-chains (*Pegg, 2009b*; *Bale and Ealick, 2010*)(*Figure 2A*). Trypanosomatid AdoMetDC undergoes this cleavage, while the corresponding pseudoenzyme, which we call 'prozyme', lacks key residues and is not processed to the active form (*Willert and Phillips, 2012*). Prozyme is only found in the trypanosomatids. The *T. brucei* AdoMetDC/prozyme complex is a heterodimer, whereas *T. brucei* AdoMetDC (*Tb*AdoMetDC) alone forms a homodimer only at high concentrations (*Velez et al., 2013*; *Willert et al., 2007*). Mutagenesis and biochemical data implicated the *Tb*AdoMetDC N-terminus in the prozyme-induced activation mechanism, but the structural basis for the regulation was not elucidated (*Velez et al., 2013*).

The polyamine biosynthetic pathway is highly regulated in most eukaryotic cells (*Pegg, 2009a*). However, these regulatory mechanisms are not conserved in trypanosomatids, and instead in *T. brucei* the pathway is regulated by prozyme through both allosteric (enzyme activation upon heterodimer formation) and protein expression effects that modulate the concentration of active AdoMetDC in the cell (*Willert and Phillips, 2008*; *Xiao et al., 2013*). Some evidence for allosteric regulation has also been reported for human AdoMetDC where the polyamine putrescine has been shown to stimulate both processing to form the pyruvoyl cofactor and also enzyme activity

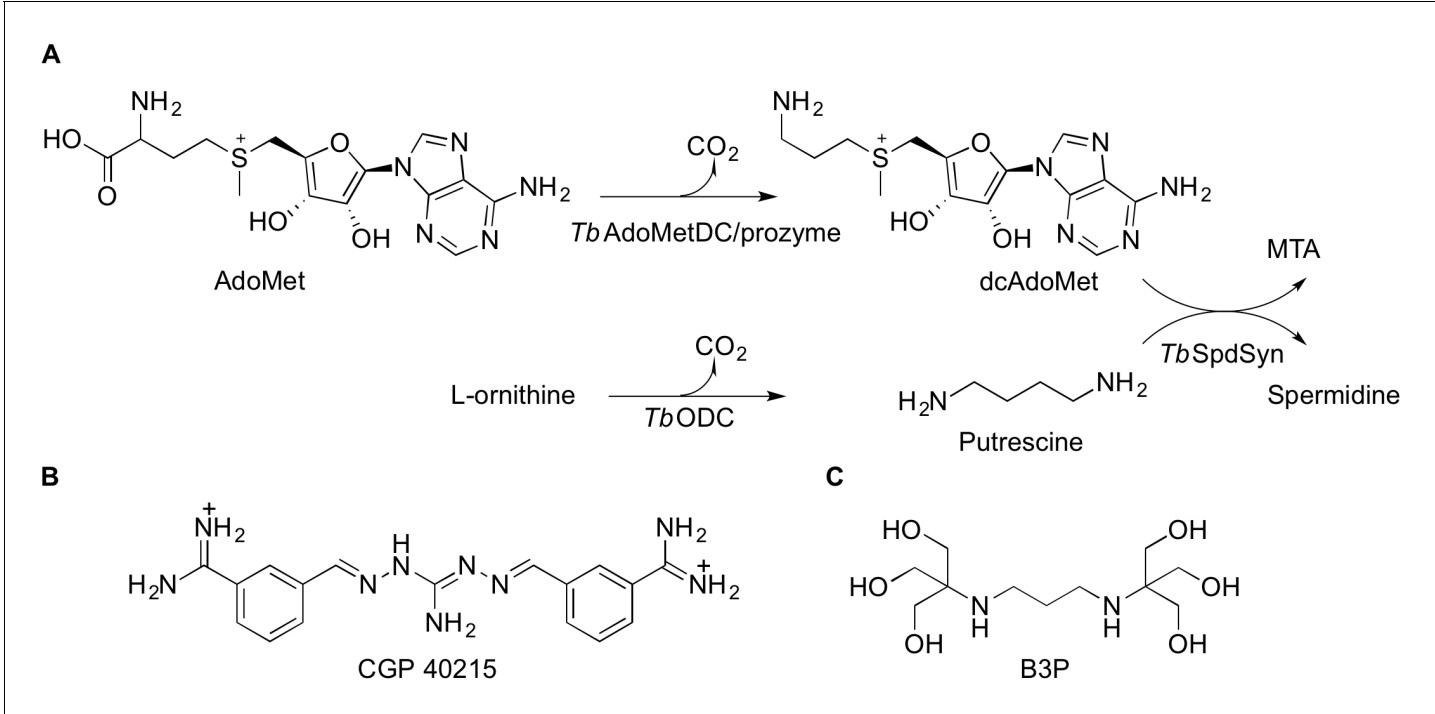

**Figure 1.** Polyamine biosynthetic pathway and *Tb*AdoMetDC ligands. (A) Reactions catalyzed by *T. brucei* S-adenosyl-L-methionine decarboxylase (*Tb*AdoMetDC/prozyme heterodimer), ornithine decarboxylase (*Tb*ODC) and spermidine synthase (*Tb*SpdSyn) are shown. AdoMet, *S*-adenosyl-L-methionine; dcAdoMet, decarboxylated *S*-adenosyl-L-methionine; MTA, methylthioadenosine. (B) CGP 40215 (CGP) is a competitive inhibitor of AdoMetDC (C) Bis-tris propane (B3P), a buffer component in the *Tb*AdoMetDC/prozyme crystallization solution.

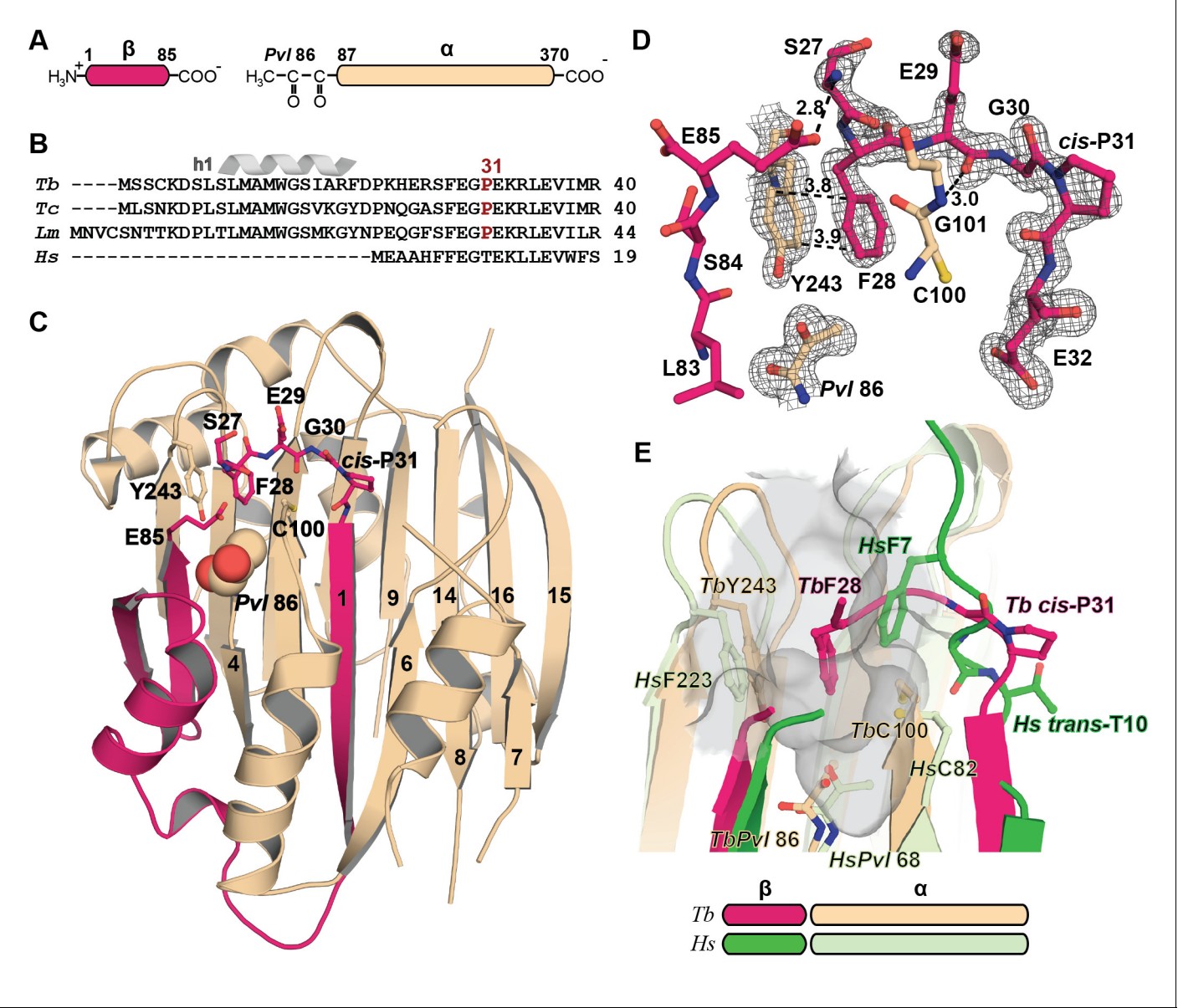

**Figure 2.** Mechanistic basis for the inactivity of the *Tb*AdoMetDC△26 monomer. (**A**) Schematic representation of *Tb*AdoMetDC β-(pink) and α-(beige) chains resulting from autocatalytic serinolysis and pyruvoyl (*Pvl*86) formation. (**B**) Sequence alignment of the N-termini of trypanosomatid (*Tb*, *T. brucei*; *Tc*, *T. cruzi*; and *Lm*, *Leishmania major*) and *Hs*, human, AdoMetDCs. For the complete sequence alignment see ***Supplementary file 2***. (**C**) Ribbon diagram of *Tb*AdoMetDC△26 (β-chain in pink and α-chain in beige). Select β-strands are numbered. *Pvl*86 is shown in spheres and select residues in the autoinhibitory sequence and active site as sticks. Atom colors follow standard nomenclature where carbon is the indicated color, nitrogen (blue), oxygen (red) and sulfur (yellow). (**D**) 2m|F$_o$−DF$_c$| electron density map of autoinhibitory residues contoured at the 1.2 σ. Dashed lines indicate distances (Å). (**E**) Active site comparison of *Tb*AdoMetDC△26 and *Hs*AdoMetDC (3DZ6) (β-chain dark green, α-chain light green). *Hs*AdoMetDC active site surface is gray. Structures aligned with an RMSD of 2.6 Å over 277 C$_\alpha$ atoms. For full structural alignment see ***Figure 2—figure supplement 1***.

The following figure supplement is available for figure 2:

**Figure supplement 1.** Comparison of *T. brucei* and human (*Hs*) AdoMetDC.

(*Pegg, 2009b*; *Bale and Ealick, 2010*). Putrescine also stimulates the activity of monomeric *Tb*Ado-MetDC, though the effects are small in comparison to prozyme (10-fold versus 1000-fold) (*Velez et al., 2013*; *Willert et al., 2007*). How these various regulatory strategies evolved in the AdoMetDC family and how they are related remain open questions.

In order to determine the structural basis for prozyme activation of *Tb*AdoMetDC, and to assess the contribution of metabolites such as putrescine to the mechanism, we solved atomic resolution X-ray structures of the *Tb*AdoMetDC inactive monomer and of the active *Tb*AdoMetDC/prozyme heterodimer. The structures show that the inactive monomer is autoinhibited by residues in the N-terminus and that a coupled set of conformational changes are required to relieve the autoinhibition leading to *Tb*AdoMetDC activation by prozyme. These structural movements include a *cis*-to-*trans* proline isomerization that facilitates positioning of the N-terminal α-helix into the heterodimer interface and an accompanying β-sheet reorganization. Our studies provide an example of how a pseudoenzyme can mediate allosteric control of enzyme activity through a mechanism involving multiple distributed conformational changes. Interestingly, comparative analysis of the AdoMetDC family shows that structural features responsible for this mechanism are present in other eukaryotic AdoMetDC enzymes, providing a model for how such a complex cooperative mechanism could arise through a process of stepwise variation and selection.

## Results

### X-ray structure determination of the inactive T. brucei AdoMetDC monomer

Trypanosomatid AdoMetDCs contain a highly conserved 16 amino acid N-terminal sequence that is not conserved in other species (*Figure 2B* and *Supplementary file 2*). Deletion or mutation of these residues led to loss of prozyme-mediated activation of *Tb*AdoMetDC despite competent hetero-dimer formation implicating the N-terminus in the activation mechanism (*Velez et al., 2013*). To facilitate crystallization, we created an additional deletion mutant *Tb*AdoMetDCΔ26. Like the Δ16 mutant, *Tb*AdoMetDCΔ26 had low activity either as a monomer or after heterodimerization with prozyme, and both were modestly (3-fold) more active in the presence of putrescine (*Table 1*). *Tb*AdoMetDCΔ26 yielded crystals (space group P2$_1$2$_1$2$_1$) that diffracted to 1.48 Å resolution (*Table 2*). A single *Tb*AdoMetDCΔ26 αβ-monomer was found in the asymmetric unit composed of a four-layer sandwich with two central β-sheets positioned between outer α-helices (*Figure 2C*). The N-terminal β-sheet (strands β1-β8) contains pyruvoyl at the N-terminus of β4 (pyruvoyl β-sheet) and is composed of residues from both the α- and β-chains, whereas the C-terminal β-sheet (scaffolding β-sheet) contains only α-chain residues (strands β9-β16) (*Figure 2C*). To assess the contribution of putrescine to the *Tb*AdoMetDC activation mechanism it was included in crystallization buffers.

**Table 1.** Activity of *T. brucei* AdoMetDC and AdoMetDC/prozyme complexes.

| | | $k_{cat}$/K$_m$ (s$^{-1}$M$^{-1}$) | | | |
|---|---|---|---|---|---|
| | | *Tb*AdoMetDC monomer | | *Tb*AdoMetDC/prozyme heterodimer | |
| *Tb*AdoMetDC proteins | Prozyme proteins | (+) Put | (−) Put | (+) Put | (−) Put |
| Wild-type | Wild-type | 9.7 ± 3.5 | 0.48 ± 0.08 | 3.2 ± 0.6×10$^3$ | 2.6 ± 0.3×10$^3$ |
| Δ16 | Wild-type | 16±3[*] | 16±11[*] | 28±5[*] | 18±3[*] |
| Δ26 | Wild-type | 0.32 ± 0.07 | 0.11 ± 0.02 | 7.6 ± 1.7 | 2.2 ± 0.3 |
| H172A | Wild-type | ND | ND | 2.1 ± 0.4×10$^3$ | 93 ± 42 |
| W137A/M146A | Wild-type | ND | ND | 87 ± 8 | 5.0 ± 0.9 |
| Wild-type | M148'A/Y152'A | as above | as above | 0.66 ± 0.23×10$^3$ | 16 ± 4 |

[*] data taken from (*Velez et al., 2013*). Data were collected in the presence of 4 mM putrescine (Put), except for the Δ16 mutant (5 mM putrescine[*]). In all cases, the heterodimer with the wild-type counterpart formed with sufficient affinity that the subunits could be copurified as a stable complex. Error represents the standard deviation for the fit of triplicate data points. ND, not determined.

**Table 2.** Crystallographic diffraction data and refinement statistics.

| | _Tb_AdoMetDCΔ26 monomer | _Tb_AdoMetDC/prozyme heterodimer | _Tb_AdoMetDC/prozyme heterodimer with CGP |
|---|---|---|---|
| **Data collection** | | | |
| Space group | P2₁2₁2₁ (No. 19) | P2₁ (No. 4) | P2₁ (No. 4) |
| Unit cell dimensions (Å, °) | a = 46.67, b = 75.64, c = 87.92 | a = 81.30, b = 96.71, c = 99.58; β = 102.64 | a = 81.13, b = 96.31, c = 98.48; β = 102.22 |
| Wavelength (Å) | 0.97935 | 0.97932 | 0.97934 |
| Average mosaicity (°) | 0.25 | 0.85 | 0.62 |
| Resolution range (Å) | 50–1.48 (1.51–1.48)[*] | 50–2.40 (2.44–2.40) | 50–2.42 (2.46–2.42) |
| Unique number of reflections | 51,575 | 57,780 | 57,137 |
| Average redundancy | 9.4 (4.8) | 6.9 (5.6) | 4.0 (3.4) |
| Completeness (%) | 98.2 (79.2) | 99.5 (97.4) | 99.4 (98.5) |
| $R_{r.i.m.}$ (%) [‡] | 6.8 | 14.6 | 12.5 |
| $R_{p.i.m.}$ (%) [§] | 2.2 (39.3) | 5.5 (60.2) | 6.1 (69.7) |
| $<I/\sigma_I>$ | 29.8 (1.4) | 15.1 (1.3) | 16.6 (1.7) |
| $CC_{1/2}$ in the last resolution shell | 0.68 | 0.60 | 0.53 |
| CC* in the last resolution shell | 0.90 | 0.87 | 0.83 |
| Wilson $B$-factor (Å²) [†] | 17.2 | 42.0 | 25.1 |
| **Refinement** | | | |
| Resolution range (Å) | 36.2–1.48 (1.53–1.48) | 37.9–2.41 (2.49–2.41) | 32.4–2.42 (2.51–2.42) |
| Number of reflections Total/$R_{free}$ | 51,476/2000 (4315/167) | 57,703/2000 (5361/186) | 55,801/1674 (5172/155) |
| Atoms (non-H protein/ligands/solvent) | 2584/6/252 | 10,275/60/101 | 10,214/124/261 |
| Protein residues (resolved/sequence) | 314/344 [¶] | 1292/1390 [¶,**] | 1282/1390 [¶,**] |
| $R_{work}$ (%) | 15.8 (25.1) | 22.8 (32.6) | 21.3 (29.1) |
| $R_{free}$ (%) | 20.0 (32.6) | 27.1 (33.2) | 25.5 (34.6) |
| RMSD bond length (Å) | 0.008 | 0.002 | 0.002 |
| RMSD bond angle (°) | 1.0 | 0.46 | 0.48 |
| Average B-factor (Å²) (protein/ligands/solvent) | 24.9/20.8/32.3 | 56.2/56.3/40.2 | 40.6/56.2/28.9 |
| Ramachandran plot (%) (favored/allowed/disallowed) | 98.1/1.6/0.3 [††] | 96.4/3.6/0 | 96.9/3.1/0 |
| Poor rotamers (%) | 0.34 | 0.70 | 0.18 |
| Clashscore | 1.18 | 1.23 | 1.38 |

[*] Numbers in parentheses correspond to the last resolution shell.

[†] Maximum likelihood estimate of the overall _B_-value reported in _Phenix_.

[‡] Redundancy-independent merging R factor, $R_{r.i.m} = \sum_{hkl}\{N(hkl)/[N(hkl)-1]\}^{1/2} \times \sum_i |I_{i(hkl)} - \langle I(hkl)\rangle| / \sum_{hkl}\sum_i I_i(hkl)$ (**Weiss, 2001**)

[§] Precision-indicating merging R factor, $R_{p.i.m} = \sum_{hkl}\{1/[N(hkl)-1]\}^{1/2} \times \sum_i |I_{i(hkl)} - \langle I(hkl)\rangle| / \sum_{hkl}\sum_i I_i(hkl)$ (**Weiss, 2001**)

[¶] Residue count includes Pvl but [**] excludes the first Ser after Ulp1 cleavage site.

[††] I168 is the only residue in the disallowed region of the Ramachandran plot.

However, no density consistent with a bound putrescine was observed in the structure of _Tb_AdoMetDCΔ26.

## An autoinhibitory sequence blocks the active site of the T. brucei AdoMetDC monomer

The inactivity of _Tb_AdoMetDCΔ26 is explained by an autoinhibitory mechanism mediated by residues S27-G30, which block the active site (**Figure 2C–E**). The G30-P31 peptide bond is in the _cis_-

conformation, orienting S27-G30 into the active site. This configuration places F28 within 3.8 Å of the pyruvoyl overlapping with the active site of the previously reported *Hs*AdoMetDC structure (*Ekstrom et al., 1999*) (*Figure 2E* and *Figure 2—figure supplement 1*). In contrast, the equivalent peptide bond in *Hs*AdoMetDC (*Hs*G9-T10) is in the *trans*-conformation, and the N-terminus extends away from the active site, positioning *Hs*F7 (equivalent to *Tb*F28) to form one wall of the substrate-binding site (*Figure 2E* and *Figure 2—figure supplement 1*). The *Tb*AdoMetDCΔ26 autoinhibitory residues are well defined by electron density and form extensive interactions in the active site stabilizing the observed conformation (*Figure 2D*). These interactions include, π-π-stacking between F28 and Y243 and H-bonds between the side chain of E85 and the backbone amide of S27, and between the amide of G101 and the carbonyl of E29. P31 is conserved throughout the trypanosomatid AdoMetDCs (*Figure 2B* and *Supplementary file 2*). Taken together with the specificity of the observed interactions between the autoinhibitory sequence and the active site, these data strongly support the conclusion that the autoinhibition observed for the truncated monomer will also form the structural basis for inactivity of the full-length *Tb*AdoMetDC monomer.

## X-ray structure determination of the T. brucei active AdoMetDC/prozyme heterodimer

*Tb*AdoMetDC/prozyme wild-type heterodimer was crystallized with putrescine both in the absence (apo-*Tb*AdoMetDC/prozyme) and presence (CGP-*Tb*AdoMetDC/prozyme) of CGP 40215 (CGP), a known AdoMetDC inhibitor (*Bacchi et al., 1996*) (*Figure 1*). Crystals (space group P2₁) from both conditions diffracted to 2.4 Å (*Table 2*). Two *Tb*AdoMetDC/prozyme heterodimers are observed per asymmetric unit. They are associated in a tetrameric structure formed partially through a domain swap involving prozyme β-strands (β0'), but which buries a relatively small surface area (*Figure 3—figure supplement 1*). The *Tb*AdoMetDC/prozyme complex was previously shown to be a dimer in solution by analytical ultracentrifugation (*Velez et al., 2013*; *Willert et al., 2007*), thus the tetramer is unlikely to be relevant to its catalytic function. Both apo- and CGP-*Tb*AdoMetDC/prozyme structures contain ligands bound at identical sites between the β-sheets (*Figure 3A* and *Figure 3—figure supplement 2*). In prozyme, the electron density is consistent with a bound putrescine (Put') (*Figure 3—figure supplement 2C*); however, in *Tb*AdoMetDC the corresponding electron density is larger and the crystallization buffer bis-tris propane (B3P) was modeled into the site (*Figures 1* and *3A*, and *Figure 3—figure supplement 2A*). The CGP-*Tb*AdoMetDC/prozyme structure additionally has CGP bound in the active site and a second putrescine (Put) found in a novel site near the N-terminal α-helix (h1) (*Figure 3A* and *Figure 3—figure supplement 2*). The ligand binding sites are described in greater detail below.

Prozyme and *Tb*AdoMetDC subunits share the same overall fold (*Figure 3B,C*). However, besides the pyruvoyl, prozyme is also missing additional active site residues including the ligand-binding residue Y243, helices h9-h11 and strand β10. On the dimer interface side of the subunits, h6 and h7, which form part of the h1 binding site, and h8 in *Tb*AdoMetDC are composed of only a single helix (h8') in prozyme and the orientation of these helices is also substantially different (*Figure 3B,C*).

The *Tb*AdoMetDC/prozyme dimer interface contains an extensive buried surface area (3,200 Å²). Strands β9-β16 form a scaffold that extends across the dimer interface with H-bond interactions formed between β15 from *Tb*AdoMetDC (R336-E340) and β15' from prozyme (R305-H309) (*Figure 3C,D*). Additional interface interactions are observed between AdoMetDC pyruvoyl β-sheet (β7 and h8) and prozyme loop β7'-β8', and vice versa.

## Structural basis for activation of TbAdoMetDC by prozyme

Comparison of inactive *Tb*AdoMetDCΔ26 with *Tb*AdoMetDC in the active heterodimer complex reveals a coupled set of conformational changes explaining how heterodimerization with prozyme leads to enzyme activation. Heterodimerization is likely initiated by the formation of the extended β-sheet across the dimer interface between the structurally rigid scaffolding β-sheet of *Tb*AdoMetDC and its counterpart in prozyme (*Figure 3D*), providing a platform to support the conformational changes required for activation. The structural reorganization is defined by three segments of movement: (1) repositioning of the autoinhibitory residue and N-terminal h1 helix, (2) reorganization of the pyruvoyl β-sheet, and (3) ordering of the β6-h8 connector loop (*Figure 4* and *Video 1*).

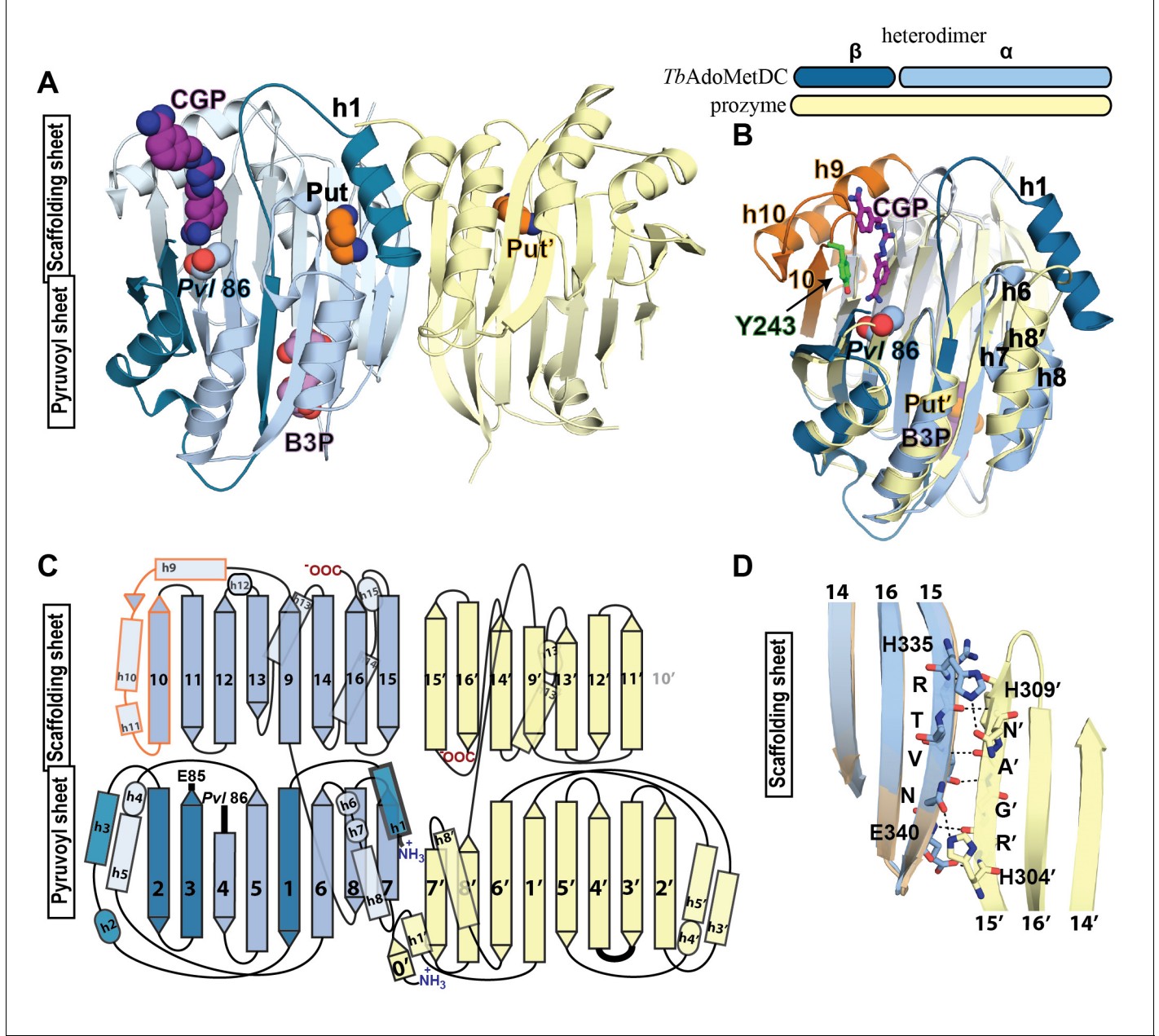

**Figure 3.** Structural organization of the *Tb*AdoMetDC/prozyme heterodimer. (**A**) Ribbon diagram of the CGP-bound heterodimer of *Tb*AdoMetDC (teal (β) and sky blue (α)) and prozyme (yellow). The schematic depicts color codes for the various chains. Ligand colors are as follows: *Pvl*86 (sky blue), CGP (purple), Put (orange)(AdoMetDC site), Put' (orange)(prozyme site), and B3P (violet) are shown as spheres. Residues and ligands in prozyme are marked ('). (**B**) Superposition of *Tb*AdoMetDC and prozyme subunits from the CGP-heterodimer (RMSD = 2.4 Å over 261 Cα atoms). *Tb*AdoMetDC active site helices/strands (residues 202–247) absent from prozyme are colored orange, Y243 (green), CGP (purple), B3P (violet), and prozyme Put' (orange). (**C**) Schematic representation of the *Tb*AdoMetDC/prozyme heterodimer. Prozyme helices (rectangles) and strands (arrows) were numbered based on structural homology to *Tb*AdoMetDC. (**D**) Superposition of the scaffolding sheets from *Tb*AdoMetDC△26 and the apo-*Tb*AdoMetDC heterodimer subunit showing main- and side-chain H-bond network (dashed lines) across the dimer interface (overall structures RMSD = 2.1 Å over 310 Cα atoms). For the tetramer structure observed in the asymmetric unit see *Figure 3—figure supplement 1* and for the electron density supporting ligand placement see *Figure 3—figure supplement 2*.

The following figure supplements are available for figure 3:

**Figure supplement 1.** *Tb*AdoMetDC/prozyme tetrameric structure.

**Figure supplement 2.** Simulated-annealing composite omit map around ligands in CGP-*Tb*AdoMetDC/prozyme heterodimer.

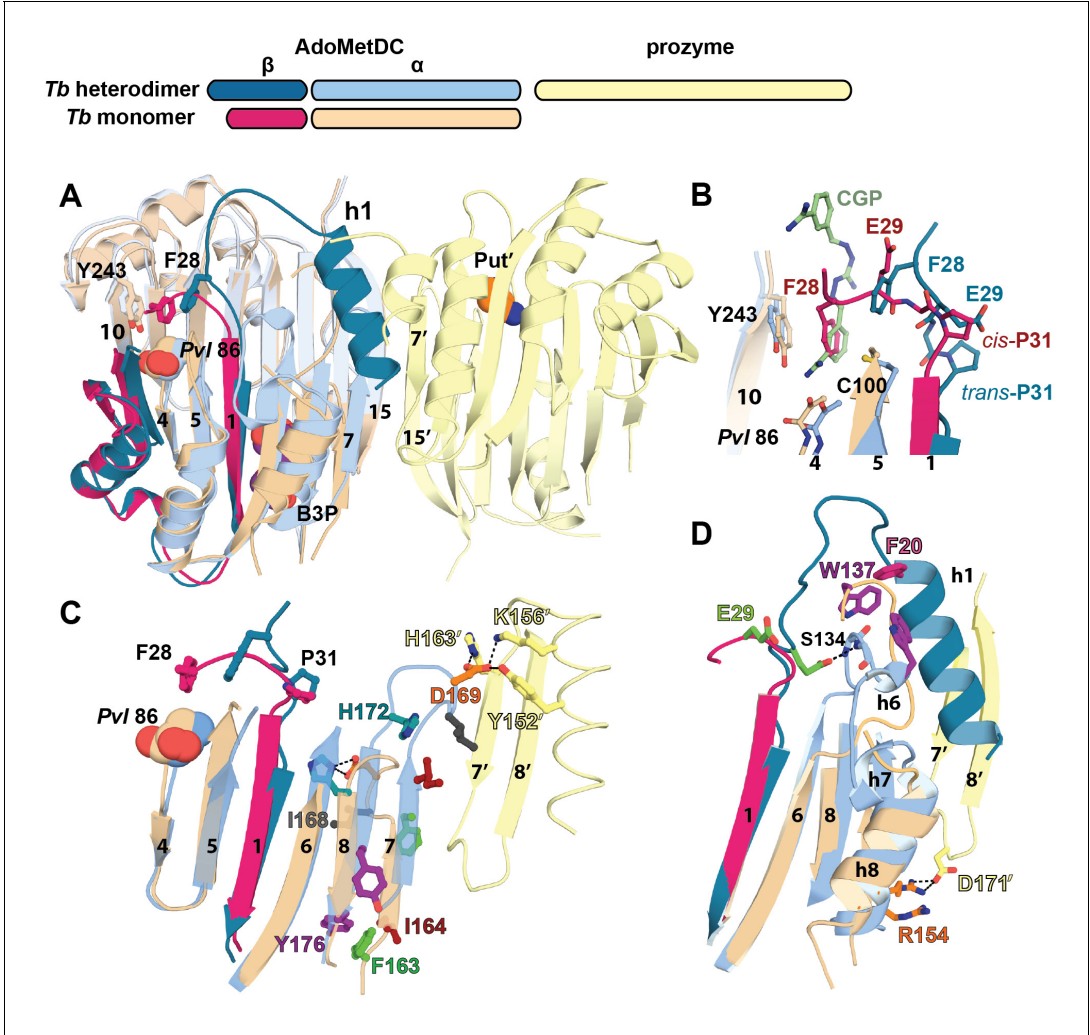

**Figure 4.** Structural basis for allosteric activation of *Tb*AdoMetDC by prozyme. (**A**) Ribbon diagram of superimposed inactive *Tb*AdoMetDCΔ26 (pink/beige) with active apo-*Tb*AdoMetDC/prozyme (teal/sky blue/yellow) (RMSD = 2.1 Å over 310 $C_\alpha$ atoms). The schematic depicts color codes for the various chains. Ligands are colored as follows: B3P (purple) and Put′ (orange). (**B**) Active site comparison of inactive *Tb*AdoMetDCΔ26 with CGP-*Tb*AdoMetDC/prozyme. Inhibitor CGP is shown in green. (**C**) Pyruvoyl β-sheet rearrangements between inactive *Tb*AdoMetDC△26 and the apo-*Tb*AdoMetDC subunit from the heterodimer. Representative residues on the β7 and β8 strands and nearby loops are highlighted as color-coded pairs: Y176 (purple); F163 (green); I164 (red); I168 (black); H172 (teal) and D169 (orange). Labels are positioned near the monomer for F163, I164, I168, and Y176 and the dimer for D169 and H172. Highlighted residues migrate over distances in parenthesis ($C_\alpha$-to-$C_\alpha$) between the inactive monomer and heterodimer structures: F163 (14.7 Å), I164 (15.2 Å), I168 (13.7 Å), D169 (14.3 Å), H172 (8.8 Å), Y176 (8.4 Å). (**D**) β6 to h8 connector (residues 130–145) rearrangements between the inactive monomer and the active heterodimer. Residues E29 (green) (5.0 Å), W137 (purple) (15.4 Å), R154 (orange) (3.5 Å) are shown as color coded pairs and the migration distances are in parenthesis. F20 is shown in pink and all other residues are colored the same as their chain color. For a schematic and surface representation of these conformational changes see *Figure 4—figure supplement 1* and *Figure 4—figure supplement 2*, respectively. See also *Video 1*.

The following figure supplements are available for figure 4:

**Figure supplement 1.** Diagram of β-sheet rearrangement between the inactive *Tb*AdoMetDCΔ26 monomer and the active *Tb*AdoMetDC/prozyme heterodimer.

**Figure supplement 2.** Comparative surface diagram of amino acid residue movement between the inactive *Tb*AdoMetDCΔ26 monomer and the active CGP-*Tb*AdoMetDC/prozyme heterodimer.

Firstly, in the *Tb*AdoMetDC/prozyme heterodimer, the autoinhibitory residues (S27-G30) have been displaced from the active site, which relieves the autoinhibition and positions F28 to form the catalytically competent ligand-binding site (*Figure 4A,B*). This movement is facilitated by *cis*-to-*trans* isomerization of P31. The *Tb*AdoMetDC N-terminal α-helix (h1, residues S7-R19) then docks into the heterodimer interface stabilizing the open conformation of the active site (*Figures 3A*, *4A* and *5A*).

Secondly, the formation of the h1 binding site is accompanied by rearrangement of the pyruvoyl β-sheet (*Figure 4C*). While the scaffolding β-sheets and connecting coils and helices (*Tb*AdoMetDC residues *Tb*189-356) align closely between the heterodimer and monomer structures (RMSD of 1.1 Å for 168 equivalent $C_\alpha$ atoms), the β-chain and α-chain residues *Tb*87-179 forming the pyruvoyl β-sheet and connecting coils and helices show significant deviations (RMSD of 5.1 Å for 145 equivalent $C_\alpha$ atoms) (*Figure 4C,D*). The two *Tb*AdoMetDC β-strands (β7 and β8) closest to the dimer interface show the most significant movement undergoing a nested set of β-strand slips. As a unit, β7 and β8 experience a shift in H-bond register of 3 residues relative to β6 while additionally slipping by 2 residues (in-register) relative to each other (*Figure 4C*, *Figure 4—figure supplement 1* and *Video 1*). Relative to β6, the position of β7, β8 and connecting loop translates towards the dimer interface, while β7 and β8 flip to reposition the side chains from one surface to the other. In the heterodimer residues on β7 (F163, F165) and β8 (Y173, L175) have flipped away from the h1 binding site to pack against the scaffolding β-sheet, whereas residues on the other face (β7: I164, I168 and β8:H172, F174, Y176) have reoriented towards h1. The two residue slip between β7 and β8 leads to an elongation of the β7-β8 connecting loop so that the combined effect is that residues in these strands and connecting loops undergo 8–15 Å migrations between the monomer and heterodimeric structures. The elongated loop (G166-H172) forms direct interactions with prozyme across the interface including a salt bridge between *Tb*AdoMetDC D169 and prozyme residues (K156' and H163') and an H-bond with Y152' (*Figure 4C*). The reorganization of β7 and β8 propagates across the sheet leading to reorientation of the catalytic base C100 (strand β5) in the active site (*Figure 4B*). The analogous residues in *Hs*AdoMetDC (C82) and in *T. cruzi* AdoMetDC were previously demonstrated to function as general acids during catalysis (*Kinch and Phillips, 2000*; *Xiong et al., 1999*).

In the *Tb*AdoMetDC monomer structure the backbone torsion angels for I168 are in the disallowed region of the Ramachandran plot, suggesting the strained conformation may help to promote the structural rearrangements. I168 is part of a trypanosomatid AdoMetDC-specific conserved sequence motif ($^{168}$I/VDSDHY$^{173}$)(*Supplementary file 2*) that also contains H172 (novel putrescine binding-site) and D169 (interacts across the dimer interface with prozyme), consistent with its involvement in the activation mechanism. However I168 forms lattice contacts in the crystal, thus we cannot rule out the possibility that the disallowed conformation results from a lattice effect.

Finally, the formation of the h1 binding site requires that the *Tb*AdoMetDC loop between β6 and h8 (H130-E138), which sterically blocks the h1 binding pocket in the inactive monomer, be repositioned (*Figure 4D*, *Figure 4—figure supplement 2* and *Video 1*). Together with adjacent disordered residues (Q139-P142) this loop reorganizes in the heterodimer to form two short $3_{10}$-helices (h6: P136-E138 and h7: G141-L144) that align with similar helices in *Hs*AdoMetDC (*Figure 6—figure supplement 1*). Residues in the loop migrate over 15 Å and new interactions between h6 and h1 are formed. These include an edge-to-face stacking interaction between W137 and F20, while repositioning of h8 allows the formation of an H-bond between R154 and D171', likely stabilizing the dimer interface. Coupled with these changes E29 from the autoinhibitory sequence undergoes a 5 Å shift to H-bond with the backbones of S134 and F135 (*Figure 4D*).

## The h1 binding site in TbAdoMetDC/prozyme

A key feature of the mechanism is that the active conformation is stabilized by insertion of h1 helix into a largely buried pocket within the dimer interface (*Figure 5A–D*). We previously showed that residues in the conserved trypanosomatid AdoMetDC N-terminus (L8, L10, M11, and M13) contributed to activation by prozyme and that the loss in activity of these mutants could be partially restored by putrescine (*Velez et al., 2013*). Extensive interactions are formed between these N-terminal *Tb*AdoMetDC residues and amino acids in both *Tb*AdoMetDC and prozyme (*Figure 5B,D*). Furthermore, the active conformation is likely stabilized and perhaps regulated by pathway metabolites since putrescine is bound in the h1 helix pocket forming H-bonds with both h1 residues and residues forming the h1 binding pocket, such as H172 (*Figure 5C*, *Figure 3—figure supplement 2D*,

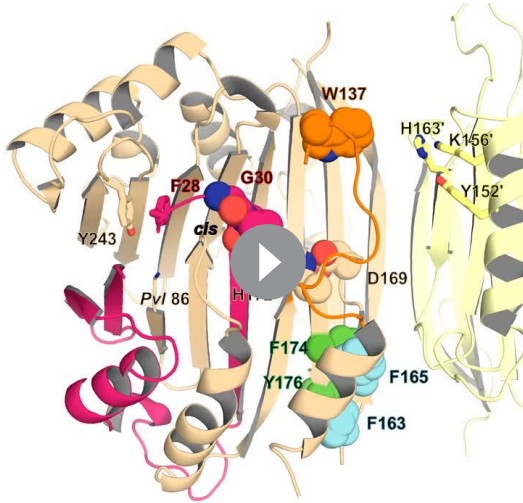

**Video 1.** Conformational rearrangements of TbAdoMetDC upon dimerization with prozyme. The movie shows the changes within *Tb*AdoMetDC from its confirmation as a *Tb*AdoMetDCΔ26 monomer to the CGP-bound TbAdoMetDC/prozyme heterodimer. *Tb*AdoMetDC is colored by chain (β in pink, α in beige, prozyme in yellow). The ribbon representation is based on the monomer secondary structure assignment. Key residues are shown: F28 (pink sticks) interacts with Y243 (beige sticks) in the monomer and forms the CGP (purple sticks) binding site in the heterodimer; D169 (beige spheres) interacts with H172 (beige spheres) in the monomer and moves 14 Å in the heterodimer to form new interactions with Y152', K156', H163' (yellow sticks); G30-P31 (pink spheres) forms *cis*-peptidyl bond in the monomer that *trans*-isomerizes in the heterodimer; W137 (orange spheres) is part of the β6-h8 connector (orange) that is partially disordered and blocks the h1 binding pocket in the monomer while repositioning and becoming structured in the heterodimer; F163 and F165 (cyan spheres) and F174, Y176 (green spheres) are residues on β7 and β8 strands, respectively, that flip from one surface of the β-strands in the monomer to the opposite surface in a heterodimer; pyruvoyl group (beige spheres) is only shown for the heterodimer. The morph and the movie were generated with PyMOL.

*E*). The putrescine site is formed upon the restructuring of residues between β6 and h8 upon heterodimerization (*Figures 3A* and *5*), thus it is not present in the inactive monomer.

A subset of the h1 helix interactions were evaluated by site-directed mutagenesis, as was the contribution of H172 (h1 putrescine binding site). *Tb*AdoMetDC-W137A/M146A and prozyme-M148'A/Y152'A mutants formed heterodimers with their wild-type counterparts, but were significantly impaired in their ability to be activated by heterodimerization (*Table 1* and *Figure 5E*). The quadruple mutant containing *Tb*AdoMetDC-W137A/M146A and prozyme-M148'A/Y152'A was further destabilized and could not be copurified as a complex. These data support a role for these residues in the prozyme activation mechanism. H172A also formed a heterodimer but in contrast to the other mutants it had near wild-type activity in the presence of putrescine (*Table 1* and *Figure 5E*). All three mutants were significantly more impaired in their ability to be activated by heterodimerization in the absence of putrescine, suggesting that putrescine plays a role in stabilizing the active conformation and was potentially an important contributor to the evolutionary path leading to the allosteric mechanism.

## β-sheet putrescine binding sites in TbAdoMetDC/prozyme

In addition to the novel putrescine binding site in the h1 pocket, we identified putrescine or putrescine analog binding sites between the β-sheets of the αββα sandwich that were occupied in both heterodimeric structures and in both *Tb*AdoMetDC and prozyme subunits. As described above the data supported placement of a buffer molecule B3P into the site in *Tb*AdoMetDC while putrescine was modeled into the prozyme site (*Figure 6A–C* and *Figure 3—figure supplement 2*). This ligand-binding site is at the identical position in the two paralogous subunits. Important conserved contacts with ligand are made by *Tb*AdoMetDC/prozyme' E36/E42', W125/W126', D189/D195', and D306/282'. However a number of amino acid residues (e.g. S187/E193' and S185/R191') differ between them and these differences enlarge the binding pocket in *Tb*AdoMetDC compared to prozyme. The binding of B3P is likely an artifact of the crystallization conditions and suggests that the larger pocket observed in *Tb*AdoMetDC has lower affinity for putrescine than the corresponding pocket in prozyme, thus allowing putrescine to be outcompeted by the buffer present in 25-fold excess over putrescine in the crystallization drop (50 mM B3P versus 2 mM putrescine).

The *Tb*AdoMetDC and prozyme B3P/putrescine binding sites are adjacent to the putrescine binding site in the human structure, suggesting a similar role in β-sheet stabilization (*Bale and Ealick, 2010*). However, the *Hs*AdoMetDC site is farther from the surface and only shares two common residues with *Tb*AdoMetDC/prozyme (*Hs*D174/*Tb*D189/prozymeD195' and *Hs*E15/*Tb*E36/prozymeE42')

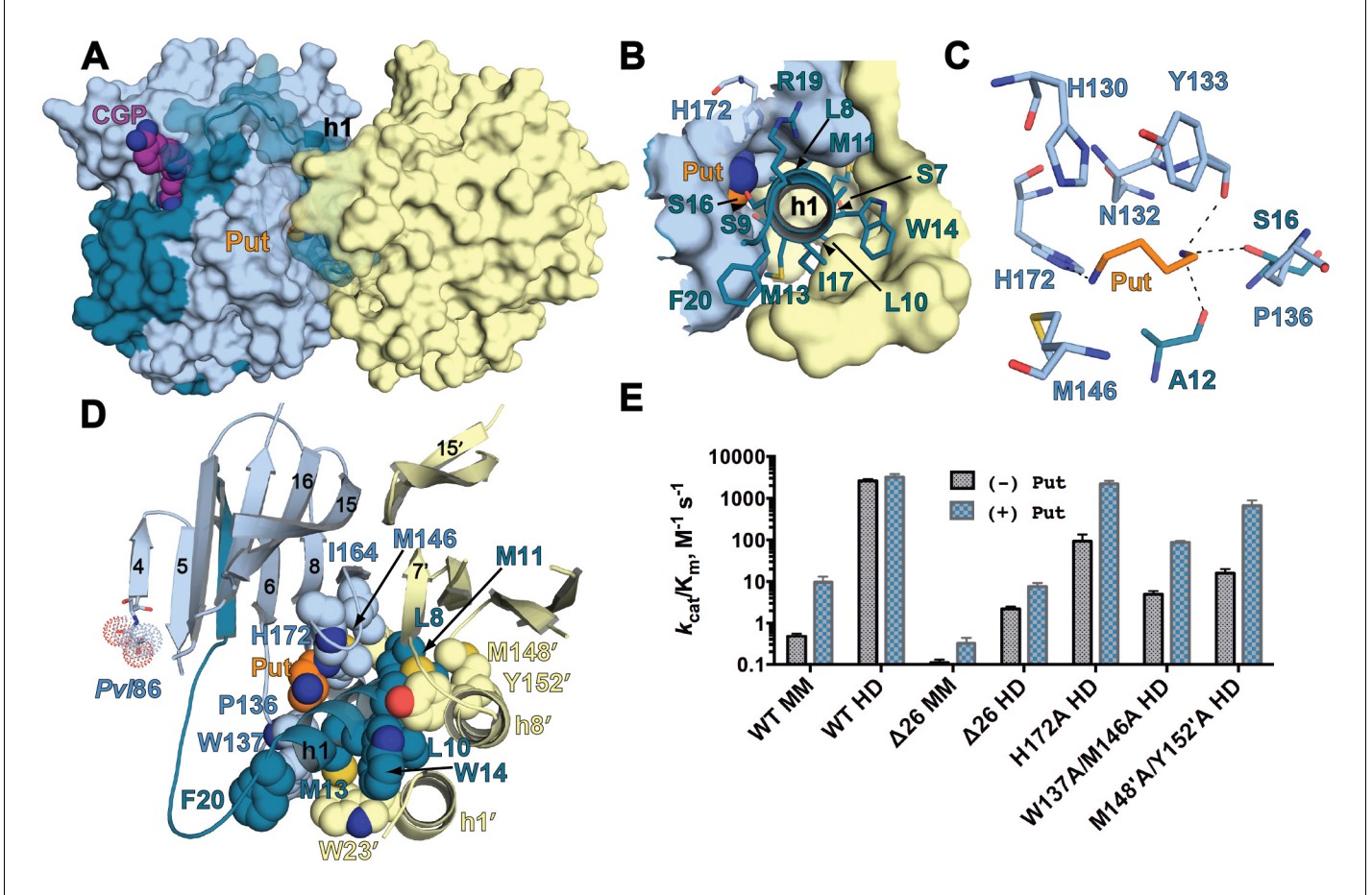

**Figure 5.** The h1 binding site in the CGP-TbAdoMetDC/prozyme structure. (**A**) Surface overview of CGP-TbAdoMetDC/prozyme. Color coding is as follows TbAdoMetDC: (teal (β) and sky blue (α)), prozyme (yellow), CGP (purple) and Put (orange). (**B**) Top view of the h1 binding site. (**C**) TbAdoMetDC h1 Put-binding site showing the 4 Å shell. Dashed lines represent H-bonds as defined by distances <3.3 Å. (**D**) Select h1 interactions (4 Å shell) with TbAdoMetDC or prozyme residues (shown as spheres). (**E**) Steady-state kinetic analysis of TbAdoMetDC and prozyme mutants for data collected ±4 mM putrescine. The enzyme rates in triplicates over the range of AdoMet concentrations used in Michaelis-Menten analysis are in *Figure 5—source data 1*.

The following source data is available for figure 5:

**Source data 1.** Enzyme rates in triplicates over the range of AdoMet concentrations used in Michaelis-Menten analysis.

(*Figure 6A* and *Supplementary file 2*). The location of the putrescine-binding site in TbAdoMetDC is supported by our previously reported mutagenesis data on *T. cruzi* AdoMetDC (low activity conformation) (*Beswick et al., 2006*; *Clyne et al., 2002*). These studies showed that D189 was a key determinant of putrescine binding and that putrescine binding could be monitored by tryptophan fluorescence, consistent with the presence of W125 in the pocket. These data support the hypothesis that the natural ligand for the TbAdoMetDC site is also putrescine (despite the presence of B3P in the pocket in our structures). However, the finding that the TbAdoMetDC binding site can accommodate a molecule significantly larger than putrescine shows it remains possible that the enzyme is regulated by binding to an unidentified metabolite.

## TbAdoMetDC/prozyme active site: CGP 40215 active site interactions

Comparison of CGP-TbAdoMetDC/prozyme heterodimer with the apo-TbAdoMetDC/prozyme structure shows that conformational changes upon ligand binding are limited to the movement of a couple of residues in the flexible connector arm (e.g. R26) adjacent to h1 and the active site

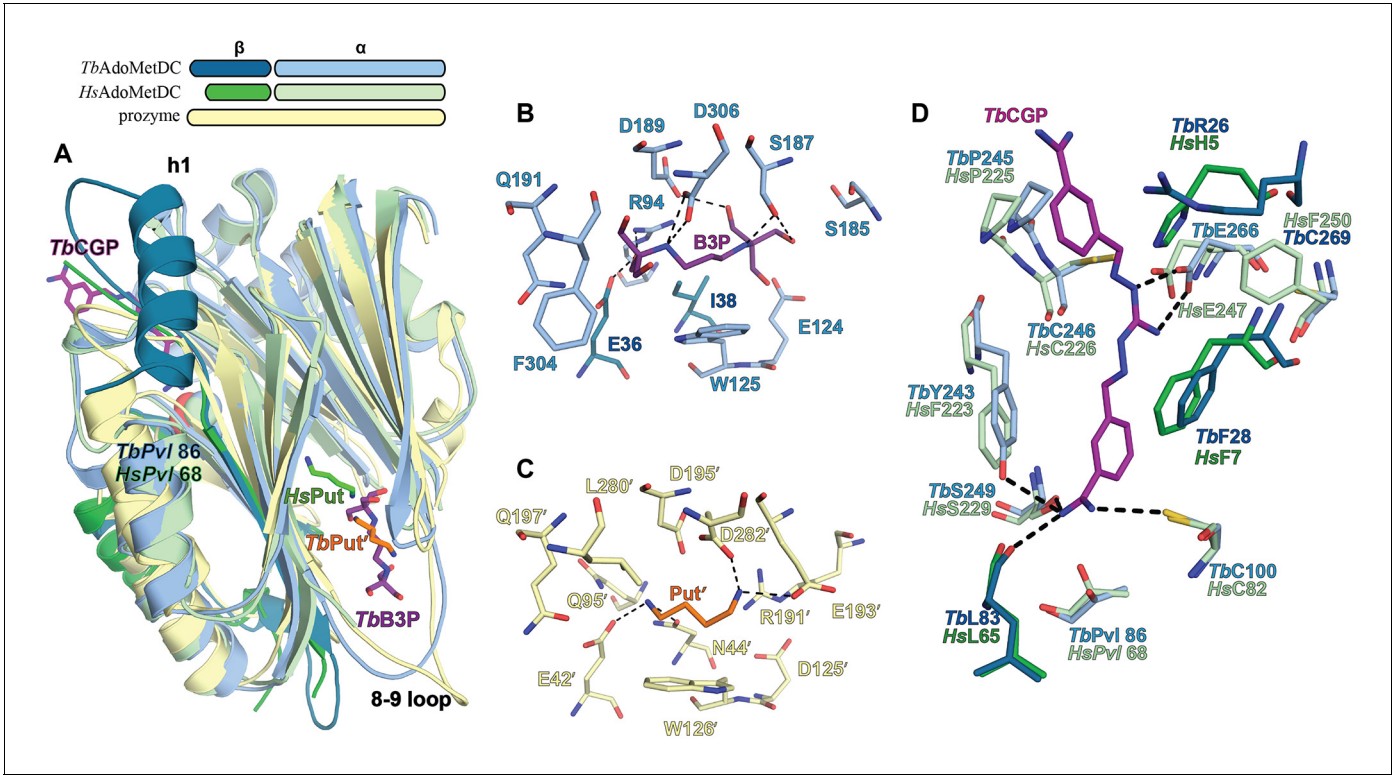

**Figure 6.** Ligand-binding sites in the CGP-*Tb*AdoMetDC/prozyme heterodimer structure. (**A**) Overlay of CGP-*Tb*AdoMetDC β/α (teal/sky blue, CGP 40215 (*Tb*CGP) and B3P in purple), prozyme (yellow, *Tb*Put' in orange) and *Hs*AdoMetDC (3DZ6) β/α (dark green/light green, *Hs*Put in green) structures viewed from the dimer interface. The schematic above the figure depicts color codes for the various chains. (**B–C**) Limited 4 Å shell showing the B3P- and putrescine-binding sites in *Tb*AdoMetDC (**B**) and prozyme (**C**). (**D**) Overlay of the *Tb*AdoMetDC CGP-binding site with *Hs*AdoMetDC showing select residues in the 4 Å shell. H-bond interactions (distance < 3.3 Å) are shown by dashed lines. The electron density supporting ligand placement is shown in *Figure 3—figure supplement 2*, the complete structural alignment of *Hs* and *Tb* AdoMetDCs in *Figure 6—figure supplement 1* and the comparison of the unliganded and liganded *Tb*AdoMetDC structures in *Figure 6—figure supplement 2*.

The following figure supplements are available for figure 6:

**Figure supplement 1.** Comparison of *Tb*AdoMetDC/prozyme and *Hs*AdoMetDC dimer.

**Figure supplement 2.** Comparison of apo- and CGP-bound *Tb*AdoMetDC/prozyme active site.

(*Figure 6—figure supplement 2*). In the presence of ligand R26 repositions to form an interaction with the π electrons in the benzamidine ring of the inhibitor. CGP binds in the *Tb*AdoMetDC active site with one amidine group buried 3.6 Å above the active site pyruvoyl group (*Figure 6D* and *Figure 3—figure supplement 2B*). CGP forms H-bonds with the backbone carbonyl of L83, the side chain hydroxyls of S249 and Y243, and with C100, suggesting C100 may be unprotonated leading to formation of an ion pair with the amidine of the inhibitor. The buried benzyl ring is sandwiched between Y243 and F28 forming π-π stacking interactions with the former and the guanidine group nitrogens (N01 and N03) form a bidentate H-bond with E266. The amidine of the second benzamidine group is solvent exposed. The fully activated *T. brucei* enzyme has a conformation and active site structure similar to the human enzyme (*Figure 6—figure supplement 1*) however two residues within the 4 Å CGP 402015 inhibitor contacting surface differ between the human and *T. brucei* structures (*Tb*R26/*Hs*H5 and *Tb*Y243/*Hs*F223) (*Figure 6D*). Additionally, substitution of *Tb*C269 for *Hs*F250 leads to a less restricted binding pocket in the *T. brucei* enzyme.

## Evidence for allosteric regulation of homodimeric AdoMetDCs from other species

The monomeric structure of eukaryotic AdoMetDC is an αββα sandwich that formed from the fusion of two smaller bacterial AdoMetDC proteins comprised of αβ half sandwiches (*Bale and Ealick, 2010*). Additionally the eukaryotic enzymes contain a dimerization domain that is not observed in the bacterial enzymes (*Figure 7*). This domain is split between the two central β-sheets suggesting it was acquired prior to the fusion of the αβ half sandwiches. Most characterized eukaryotic AdoMetDC enzymes are dimers, although the plant enzyme lost the ability to dimerize and unlike the human enzyme does not require putrescine to stimulate processing or activity (*Bale and Ealick, 2010*). Within this complex evolutionary background, the trypanosomatids underwent a gene duplication event leading to the prozyme regulatory mechanism. Though prozyme is only found in the trypanosomatids we sought to determine if the potential for AdoMetDC to be allosterically regulated arose only in the trypanosomatid lineage or whether it was an earlier invention of the AdoMetDC enzyme family. By combining structural insight into the prozyme regulatory mechanism with a phylogenetic analysis of the enzyme family, we sought to find evidence for coevolution of residues involved in the activation mechanism and thus to determine if aspects of the allosteric regulatory mechanism were conserved in other eukaryotic AdoMetDCs. Through this analysis we then hoped to be able to generate a model for how the complex allosteric control of the trypanosomatid AdoMetDC was able to evolve in a stepwise manner.

Sequence analysis of the AdoMetDC family shows that a proline residue equivalent to *Tb*AdoMetDC P31 is present in all fungal AdoMetDCs and in diverse single-celled eukaryotes including *Naegleria* (Excavata) and *Dictyostelium* (Amoebozoa) but not in animals or plants (*Supplementary file 2*). Furthermore, P31 appears to have coevolved with the presence of an extended N-terminus, relative to mammalian and plant AdoMetDCs and with several residues that play roles in the prozyme allosteric activation mechanism (*Figure 7*). These include H172 (h1 putrescine binding site), T104 (packs against the $3_{10}$-helix h6), N132 (a part of β6-h8 connector loop and h1 putrescine binding site), and C269 (within 4 Å of the autoinhibitory sequence F28). Coevolution of H172 with P31 extends throughout the fungal sequences, while T104, N132, and C269 are found in a more limited subset of fungal and protist sequences.

## Discussion

The *Tb*AdoMetDC/prozyme heterodimer structure provides insight into how an inactive pseudoenzyme can regulate its paralogous enzyme. We have shown that prozyme activates trypanosomatid AdoMetDC through an allosteric mechanism involving extensive conformational changes (*Figure 8*). *Tb*AdoMetDC is maintained in the low activity state in the absence of prozyme by autoinhibitory residues positioned in the active site by the *cis*-conformation of P31. Upon formation of the heterodimer, P31 undergoes a *cis*-to-*trans* isomerization, and helix h1 is docked into the dimer interface. This isomerization positions the autoinhibitory residues into the open configuration of the substrate-binding site leading to enzyme activation. Reorganization of the *Tb*AdoMetDC pyruvoyl β-sheet and nearby loops is required to form the h1 binding site. The buried surface that is created by the prozyme-induced conformational changes leads to structural stabilization of the alternative conformation and to expanded biological function through enzyme activation. Left unresolved is the question of whether or not these conformational changes occur by a sequential or a concerted mechanism. While our observance of the autoinhibitory sequence in the inactive monomer was in the context of an N-terminal truncation our conclusion that the inactivity of full-length monomeric *Tb*AdoMetDC is caused by auto inhibition is supported by the following observations: (1) the autoinhibitory sequence makes a number of specific interactions with residues in the active site expected to stabilize the conformation, (2) biochemical data shows that the N-terminus is necessary for the activation mechanism, (3) both P31 and the N-terminal amino acid sequence are conserved in the trypanosomatids and, (4) the highly coupled nature of the observed conformational changes links displacement of the autoinhibitory sequence to the structural reorganization that forms the h1 binding site.

Proteins that undergo structural transitions have several common features including the existence of conformers that have flexible regions or that exist in a state of diminished stability (*Bryan and Orban, 2010*). In the case of *Tb*AdoMetDC we identified several regions of the monomeric structure that are either in strained conformations or disordered, and which are likely to play key roles in

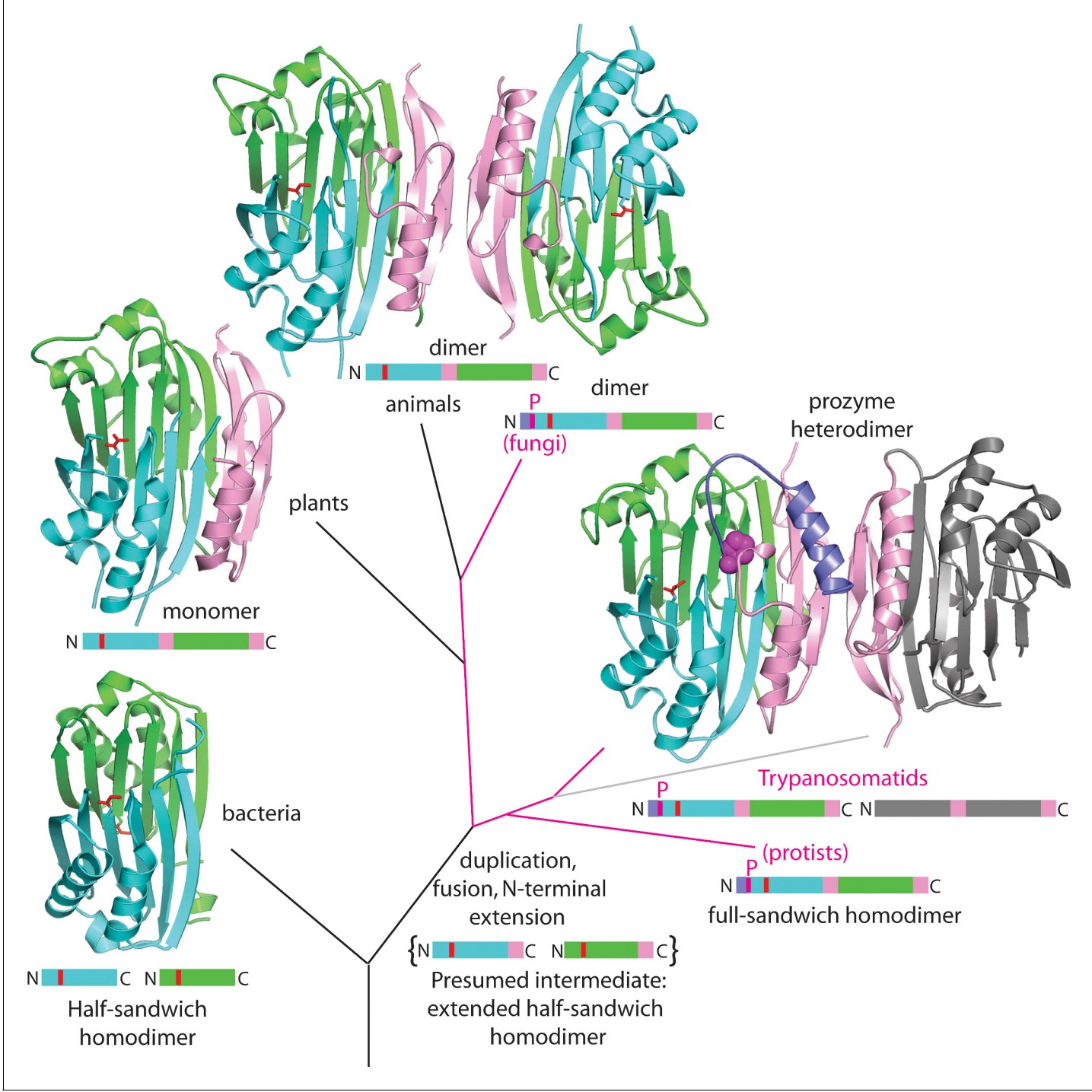

**Figure 7.** Theoretical tree diagram of the origin of eukaryotic AdoMetDCs. The diagram depicts eukaryotic AdoMetDC enzymes evolving from their bacterial counterparts by extension, gene duplication and fusion. Two bacterial half-enzymes (lower left, PDB 3iwc) form an αββα sandwich through dimerization of the β-sheet faces, with interacting chains depicted in cyan and green cartoon and active site pyruvates in red stick. The primary sequence diagram (same colors) is depicted below. The primary sequence diagram illustrates a presumed intermediate with extended C-terminal αββ extensions (pink). The duplicated and fused eukaryotic enzymes adopt the same αββα sandwich fold (now within a monomer), with extension (pink) dictating edge-to-edge dimerization of the β-sheets (*Bale and Ealick, 2010*). Trypanosomatid enzymes (represented by the *T. brucei* structure from this paper) undergo a second gene duplication, with one (prozyme in gray, lower right) losing catalytic activity. Prozyme activates the catalytic enzyme (colored cartoon, lower right) through dimerization and *cis-trans* isomerization of a conserved proline (P31, magenta sphere) with movement of N-terminal helix (purple). Additionally, protist and fungal sequences retain the conserved proline (magenta lines in the tree with primary sequence diagrams showing the location of the proline (magenta P)). Plant sequences (illustrated by PDB structure 1mhm) have lost the proline, N-terminal helix

*Figure 7 continued on next page*

Figure 7 continued

and dimerization; animal sequences (illustrated by PDB structure 1i7b) retain a dimeric structure of two active chains without proline and N-terminus (depicted in colored cartoon).

promoting the observed conformational changes. These include the *cis*-P31 peptide bond that isomerizes to form the more energetically favorable *trans*-P31, disordered surface loops that undergo a disordered-to-ordered transition, and β-strands that slip and flip during the monomer to heterodimer transition to form new favorable interactions. Within these β-strands one residue (I168) in the monomeric structure is in the disallowed region of the Ramachandran plot, suggesting that relief of the strain in the I168 backbone may also contribute energetically to the conformational reorganization. Backbone strain has previously been associated with allosteric control of catalytic activity, though has most often been found in active site residues (*Jia et al., 1993*; *Oruganty et al., 2013*). Interestingly it has also been previously noted that cis-*trans* proline isomerization is often associated with the evolution of new function when, like in the case of TbAdoMetDC, the new function uses the isomerization to drive a local conformational change (*Joseph et al., 2012*).

More generally, the mechanism of prozyme regulation of AdoMetDC provides an opportunity to explain how such intricate and complex allosteric mechanisms might evolve in a stepwise fashion. Sequence analysis shows that a proline residue equivalent to *Tb*AdoMetDC P31 and several other residues that are important for the prozyme activation mechanism are present in fungal and protist AdoMetDCs indicating that the allosteric mechanism likely arose early in eukaryotic evolution. Indeed, our analysis suggests that aspects of this regulatory mechanism may be retained in some extant fungi and single-cell eukaryotes but that it was lost in higher eukaryotes including animals and plants. As the dimerization domain is at the center of the structural rearrangements (β-strands reorganization and formation of the helix-binding site) that we observed in *T. brucei* AdoMetDC our data support the hypothesis that allostery in the eukaryotic AdoMetDC enzymes likely arose through the acquisition of this domain (*Figure 7*). The origin of the dimerization domain is not clear but it is interesting to note that human spermine synthase contains an inactive AdoMetDC domain possessing a remnant of the dimerization domain (*Wu et al., 2008*).

We propose that the dimerization domain was acquired in a single-step of variation. This event (because of high local concentration) could then immediately nucleate the further stepwise evolution of mechanisms that underlie the broad regulatory divergence of AdoMetDC in the eukaryotic lineage: regulation by metabolites, allosteric regulation by a pseudoenzyme (trypanosomatids), and simplification to produce monomeric forms (plants). Indeed, it has been proposed that novel molecular interactions (like dimerization) and then allostery can gradually evolve by a stepwise explorative process, given mechanisms that locally concentrate proteins (*Kuriyan and Eisenberg, 2007*). We further suggest that the evolution of complex allostery is facilitated by single-step acquisition of larger structural elements or domains (e.g. the AdoMetDC dimerization domain) through gene fusion events and that complex allosteric mechanisms may be unlikely to arise through point mutation alone.

Thus, the trypanosomatids likely exploited a preexisting homodimer-based regulatory mechanism, requiring only gene duplication and variation of an ancestral AdoMetDC to yield a catalytically inactive regulatory paralog. Duplication and divergence to yield prozyme also provided the basis for separate transcriptional/translational control of AdoMetDC activity adding another layer of regulation. This is particularly important in the trypanosomatids because they lack the transcriptional and translational control mechanisms used by higher eukaryotes to control polyamine biosynthesis (*Willert and Phillips, 2012*). Exploiting the two-gene organization, *T. brucei* evolved regulatory control of prozyme translation as a means to regulate AdoMetDC activity and thereby polyamine biosynthesis in the cell.

The *Tb*AdoMetDC prozyme activation mechanism shares common features with regulatory mechanisms used to control cell signaling. Autoinhibitory sequences form the basis of the inactivity of many protein kinases (*Bayliss et al., 2015*) such as JAK2 (*Zhao et al., 2009*) and EGFR (*Zhang et al., 2006*), of phospholipase C isozymes (*Gresset et al., 2012*), of GTPases (*Hansen and Kwiatkowski, 2013*) and of guanine nucleotide exchange factors (*Cherfils and Zeghouf, 2013*) such as Vav1 (*Yu et al., 2010*) and include examples of *cis*-*trans* proline isomerization to control the conformation of autoinhibitory sequences (*Craveur et al., 2013*). Odd-number β-strand slips have been

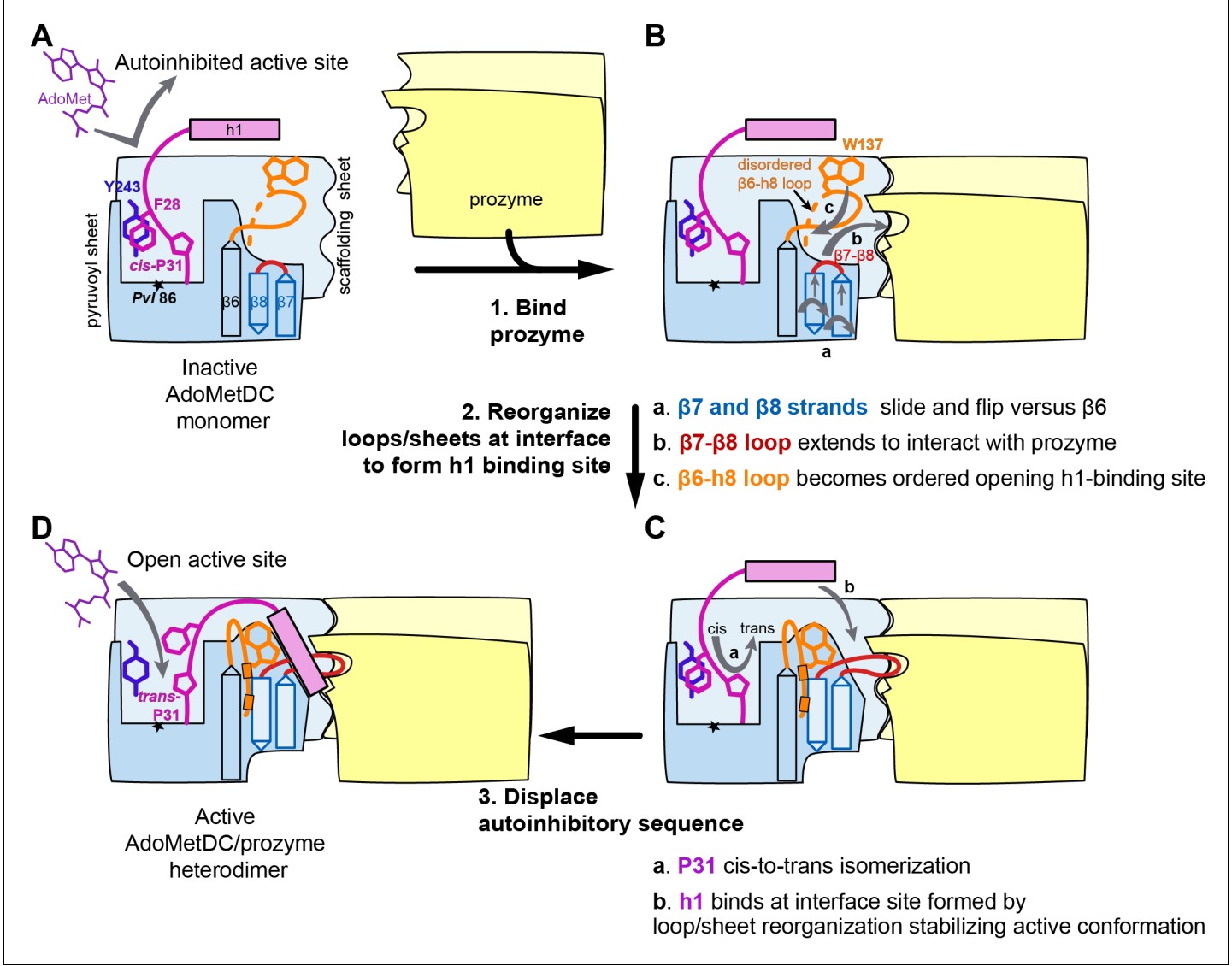

**Figure 8.** Mechanistic model of prozyme-induced *Tb*AdoMetDC enzyme activation. The model depicts a logical step-wise process assuming that the formation of the h1 binding site precedes insertion of the helix into the interface, but current data do not distinguish between a sequential versus a concerted activation mechanism and the ordering of events is hypothetical. (**A**) The inactive *Tb*AdoMetDC monomer is composed of two β-sheets: pyruvoyl (blue) and scaffolding (light blue). The active site pyruvoyl residue (*Pvl*86, star) is blocked by the inhibitory sequence (S27–G30), which is oriented into the active site by the *cis* configuration of P31. This autoinhibitory closed confirmation is stabilized by π-π stacking between F28 (purple) and Y243 (blue). Residues N-terminal of S27, including helix h1 (purple rectangle), were not present in the monomer construct and their position in the diagram is hypothetical. (**B**) Binding of prozyme (yellow/light yellow) to AdoMetDC is nucleated by formation of the H-bond network between the two scaffolding sheets leading to formation of a continuous inter-subunit β-sheet that serves as a platform for the following conformational changes: (**a**) slipping of the interface strands β7 and β8 relative to β6, which results in flipping of their side chains from one surface to the other; (**b**) repositioning and elongation of the β7-β8 loop that forms the back of the h1 binding site, stabilized in this confirmation by interaction across the interface with prozyme; and (**c**) disordered-to-ordered transition and movement of the β6-h8 loop (H130-L144, orange) that leads to formation of the h1 binding pocket. (**C**) Upon the formation of the h1 binding pocket, (**a**) *cis*-to-*trans* isomerization of P31 displaces the autoinhibitory sequence from the active site and the open active site conformation is stabilized by (**b**) docking of the h1 helix at the dimer interface. (**D**) The active *Tb*AdoMetDC/prozyme dimer is capable of binding ligands in the open active site, which leads to ~1000 fold increase in rates of AdoMet decarboxylation.

reported in a number of other allosteric systems (e.g. ARNT PAS domain [*Evans et al., 2009*]) where they orchestrate long-range conformational changes of the type observed for *Tb*AdoMetDC/pro-zyme. The finding that all of these mechanisms exist within the single AdoMetDC/prozyme

heterodimer reinforces the diversity of structural mechanisms of regulation that might originate from the formation of dimeric protein interfaces.

From a disease perspective, because of the absolute requirement for polyamines in eukaryotic cell growth, the polyamine biosynthetic pathway has been targeted for development of anti-proliferative agents. An inhibitor of the first enzyme in the pathway, ornithine decarboxylase, is used clinically for treatment for late-stage human African trypanosomiasis, suggesting other enzymes in the pathway could also be exploited for drug discovery (*Jacobs et al., 2011*; *Willert and Phillips, 2012*). We note that *Tb*AdoMetDC has a similar overall conformation to the human enzyme, but with key differences at the active site. These findings suggest that species-selective competitive inhibitors could be identified. But more intriguingly, our structures suggest the possibility of identifying species-selective inhibitors that lock the enzyme into the inactive conformation and prevent relief of autoinhibition.

The structural details of the trypanosomatid AdoMetDC regulatory mechanism may be specific to this system, but the underlying principles can be generally applicable and suggest that the prevalence of pseudoenzymes in genomes is linked to the evolution of regulatory control. Study of these other systems is likely to uncover a myriad of complex and elegant mechanisms of allosteric regulation made possible by the simple *sequential* strategy of formation of dimer interfaces, evolution of cooperative switching between functional states, gene duplication, and divergence of function. Study of pseudoenzyme regulatory mechanisms in the context of their evolution may indeed provide additional insight into how complex regulatory mechanisms arise. The evolution of the AdoMetDC/prozyme regulatory mechanism provides an exemplary case of such a process.

## Materials and methods

### Materials

General reagents and PCR primers were purchased from Sigma-Aldrich (St. Louis, MO). *S*-Adenosyl-L-methionine (AdoMet) sulfate p-toluenesulfonate salt was purchased from Affymetrix (Santa Clara, CA). *S*-Adenosyl-*L*-[carboxy-$^{14}$C]-methionine ($^{14}$C-AdoMet) was purchased from American Radiolabeled Chemicals (St. Louis, MO). Restriction enzymes and T4 DNA ligase were purchased from New England Biolabs (Ipswich, MA). *S*-adenosylmethionine decarboxylase inhibitor CGP 40215 (3-((*E*)-(((*E*)-amino(2-((*E*)−3-carbamimidoylbenzylidene)hydrazinyl)methylene)hydrazono)methyl)-benzimidamide) was a gift from Novartis (Basel, Switzerland).

### Methods

#### Generation of *E. coli* wild-type expression constructs

The *Tb*AdoMetDC open reading frame (ORF) was codon-optimized for *E. coli* and cloned into the pET28a vector by GenScript (Piscataway, New Jersey). For single-subunit expression, *Tb*AdoMetDC was cloned into the pET28bSmt3 vector, a variant of pE-SUMOpro vector (LifeSensors, Malvern, PA) as described in (*Mossessova and Lima, 2000*). The resulting construct pET28bSmt3-*Tb*AdoMetDC encoded *Tb*AdoMetDC N-terminally fused with His$_6$-tagged *Saccharomyces cerevisiae* SUMO protein (NP_010798.1), with a serine residue following the Ulp1 protease site and immediately before the first methionine residue of the *Tb*AdoMetDC sequence.

The *Tb*AdoMetDC/prozyme heterodimer coexpression construct was generated in Novagen pET-Duet-1 vector (EMD Millipore, Billerica, MA). Prozyme open reading frame (ORF) was PCR-amplified from *T. brucei* Lister 427 genomic DNA with primers p1 and p2 (primers are listed in *Supplementary file 1*) and His$_6$-SUMO-tagged *Tb*AdoMetDC ORF was amplified from pET28bSmt3-*Tb*AdoMetDC (above) with primers p3 and p4. PCR products were then sequentially cloned into the pETDuet-1 by ligation-independent cloning using the In-Fusion LIC kit (Clontech Laboratories, Mountain View, CA) per the manufacturer's instructions. The prozyme ORF was inserted first into the NdeI- and XhoI-digested pETDuet-1 vector followed by the His$_6$-SUMO-tagged *Tb*AdoMetDC ORF into NcoI/HindIII-digested pETDuet-1-prozyme construct. The resulting pETDuet-1-Smt3-*Tb*AdoMetDC-prozyme construct encoded the monocistronically transcribed tagless prozyme and gene-optimized *Tb*AdoMetDC N-terminally fused with His$_6$-SUMO.

## Site-directed mutagenesis and deletion constructs

N-terminally truncated TbAdoMetDCΔ26 was PCR amplified from the pET28bSmt3-TbAdoMetDC with primers p12 and p10 containing BamHI and XhoI endonuclease restriction sites, respectively. The amplified BamHI/XhoI-digested insert was cloned into BamH1/XhoI-digested pET28bSmt3. The resultant construct pET28bSmt3-TbAdoMetDC△26 encodes S27 of TbAdoMetDC immediately after the Ulp1 site. The TbAdoMetDCΔ26 was also cloned using BamHI-containing p12 and HindIII-containing p11 primers into BamHI/HindIII-digested pETDuet-1-Smt3-TbAdoMetDC-prozyme construct for expression of the truncated heterodimer.

The TbAdoMetDC ORF was PCR amplified from pET28b-His$_6$-Smt3-TbAdoMetDC with primers p5 and p6 and subcloned into the pCR-Blunt II-TOPO vector using Zero Blunt TOPO PCR cloning kit (Thermo Fisher Scientific, Waltham, MA). The H172A mutation was introduced by PCR using fully overlapping primers p7/p8 by QuickChange mutagenesis with PfuTurbo DNA polymerase (Agilent Technologies, Santa Clara, CA). The H172A mutant ORFs was amplified with the BamHI-containing forward primer p9 and the HindIII-containing reverse primer p11 and ligated into BamHI/HindIII-digested pETDuet-1-Smt3-TbAdoMetDC-prozyme construct for expression of the mutant heterodimer.

TbAdoMetDC-W137A/M146A and prozyme-M148A/Y152A ORFs were generated by GenScript in the context of the pETDuet-1-Smt3-TbAdoMetDC-Prozyme resulting in two constructs, each expressing a heterodimer with one of the subunits carrying a double mutation.

All DNA constructs were verified by sequencing of the TbAdoMetDC and prozyme ORFs (UT Southwestern Sanger Sequencing Core). The constructs were propagated in Invitrogen One Shot TOP10 (Thermo Fisher Scientific) or Stellar (Clontech Laboratories) cells.

## Protein purification

TbAdoMetDC, TbAdoMetDC/prozyme, and mutant enzymes were expressed from corresponding constructs in Novagen BL21(DE3)pLysS cells (EMD Millipore). Expression was induced by the addition of IPTG (0.6 mM) at OD$_{600nm}$ = 0.5–0.6, and cells were grown for 20 hr at 20°C. Cell pellets were harvested by centrifugation at 3500 x g, resuspended in buffer A (50 mM HEPES, pH 8.2, 300 mM NaCl, 10 mM imidazole, and 15% (v/v) glycerol) (15 mL per 1 L of culture) supplemented with protease inhibitors (1 µg/mL leupeptin, 2 µg/mL antipain, 10 µg/mL benzamidine, 1 µg/mL pepstatin, 1 µg/mL chymostatin, and 2 mM phenylmethylsulfonyl fluoride (PMSF)), and passed twice through the EmulsiFlex-C5 cell disruptor (Avestin, Ottawa, Canada) at 10,000 psi. Lysates were cleared by centrifugation at 56,000 x g for 3 hr at 4°C, and protein was purified by affinity chromatography using HiTrap Chelating HP columns on the ÄKTApurifier system (GE Healthcare Life Sciences, Pittsburgh, PA). After loading, the column was washed with buffer A, and protein was eluted in a gradient of imidazole from 10 to 200 mM over 25 column volumes. Protein-containing fractions were pooled and concentrated by ultrafiltration in Amicon Ultra-15 10,000 kDa NMWL centrifugal filters (EMD Millipore). To remove the His$_6$-Smt3 tag, TbAdoMetDC (both the monomer and in complex with prozyme) was incubated with His$_6$-Ulp1 (1 µg per 1 mg of protein, purified as described in [*Velez et al., 2013*]) for at least 2 hr at 4°C, sample was diluted 20-fold with buffer A and untagged protein was separated from tagged protein by passage through a HiTrap Chelating HP column equilibrated as above. Flow-through fractions were pooled and concentrated as described above. Proteins were further purified by HiLoad 16/60 Superdex 200 size-exclusion chromatography column (GE Healthcare Life Sciences) with buffer S (50 mM HEPES, pH 7.7, 50 mM NaCl) as a mobile phase. Protein purity was assessed by SDS-PAGE analysis. Protein concentrations were measured using Protein Assay Dye Reagent (Bio-Rad, Hercules, CA). Concentrated samples were flash-frozen in liquid nitrogen and stored at −80°C.

## AdoMetDC $^{14}$C enzyme activity assay

AdoMetDC activity was monitored under steady state conditions as previously described (*Willert et al., 2007*) by capturing $^{14}CO_2$ released from the $^{14}$C-AdoMet substrate onto a barium carbonate soaked filter paper enclosed in a test tube at 37°C. Assay mix contained AdoMetDC (20–40 µM) or AdoMetDC/prozyme heterodimers (0.05–15 µM), $^{14}$C-AdoMet adjusted with unlabeled AdoMet to the specific activity of 2.5 or 5 µCi/µmol (50–1600 µM total substrate concentration) in buffer (50 mM HEPES, pH 7.2, 50 mM NaCl, 4 mM DTT, 0.25 mg/mL bovine serum albumin, and

0.005% (w/v) Nonidet P-40) in a final volume of 100 µL. Reactions were run in the presence (4 mM) or absence of putrescine. Enzymes were preincubated with buffer ± putrescine at 37°C for 2 hr prior to adding substrate and initiating the assay. Preincubation did not alter activity in the absence of putrescine, but allowed for measurement of maximum activity in the presence of putrescine. Substrate dose response data were fitted to the Michaelis-Menten model in Prism (GraphPad Software, La Jolla, CA) to determine the Michaelis constant, $K_m$, and the turnover number, $k_{cat}$, and catalytic efficiency reported as $k_{cat}/K_m$. The standard deviation for each parameter was also determined by GraphPad and all fits used triplicate data for each substrate concentration.

## Protein crystallization and data collection

Crystallization of the TbAdoMetDCΔ26 monomer (10–20 mg/mL in buffer C: 50 mM HEPES, pH 7.2, 50 mM NaCl, 4 mM Tris(2-carboxyethyl)phosphine (TCEP), and 2 mM putrescine) was carried out using sitting drop vapor diffusion. Random crystallization screening was set up in 96-well CrystalMation Intelli-plates (Art Robbins Instruments, Sunnyvale, CA) on a Phoenix robotic liquid handler (Art Robbins Instruments), using equal volumes of reservoir and protein solutions with the following commercial 96-well crystallization suites: Index and PEG Rx (Hampton Research, Aliso Viejo, CA), Classics and PACT (Qiagen, Hilden, Germany), and JCSG+ and Structure1 and 2 (Molecular Dimensions, Newmarket, UK). Hexagonal rod crystals of TbAdoMetDCΔ26 appeared after two days of incubation at 20°C against a reservoir solution of 25% (w/v) PEG 3,350, 0.2 M ammonium acetate, and 0.1 M Bis-tris, pH 5.5. All crystals were flash-cooled in liquid nitrogen without additional cryoprotection.

Crystallization screening for the TbAdoMetDC/prozyme heterodimer was performed as described above using the same protein concentrations and buffers. Initial crystals were obtained by vapor diffusion against a reservoir solution containing 24–30% PEG 6000 and 0.1 M Bis-tris propane, pH 8.6–9.2. Further optimization led to growth of crystals with plate morphology against a reservoir solution of 19% PEG 6000 and 0.1 M Bis-tris propane, pH 8.0–8.6. Microseeding with these plates as source seeds (Seed-Bead, Hampton Research) diluted 50-fold with stabilization solution (50 mM HEPES, pH 7.2, 100 mM Bis-tris propane, pH 8.4, 22% (w/v) PEG 6000, 50 mM NaCl, 4 mM TCEP, 2 mM putrescine) in the presence of 4 mg/mL TbAdoMetDC/prozyme was performed to obtain single crystals of the wild-type heterodimer against a reservoir solution of 18% (w/v) PEG 6000 and 0.1 M Bis-tris propane, pH 7.9. Crystals were cryoprotected with an additional 18% (w/v) ethylene glycol, and flash-cooled in liquid nitrogen.

In order to cocrystallize TbAdoMetDC/prozyme in complex with CGP 40215 inhibitor, 4 mg/mL protein in a modified buffer C (50 mM Bis-tris propane, pH 7.2, 50 mM NaCl, 4 mM TCEP, and 2 mM putrescine) was incubated for 6 hr with 0.75 mM CGP 40215. Crystals were obtained using hanging drop vapor diffusion method set up with microseed stock solution of TbAdoMetDC/prozyme as described above. Crystals used in data collection were harvested from 17% (w/v) PEG 3,350, 0.1 M Bis-tris propane, pH 7.9, and 0.3 M NaCl, then cryoprotected with an additional 15% (w/v) ethylene glycol, and flash-cooled in liquid nitrogen.

Native diffraction data were measured at 100 K at the Structural Biology Center (Beamline 19ID) at the Argonne National Laboratory. Data were reduced using the HKL software package (*Minor et al., 2006*).

## Structure determination and model refinement

Initial phases for TbAdoMetDCΔ26 were generated by molecular replacement using the program *Phaser* (*McCoy et al., 2007*) as implemented in the program suite *Phenix* (*Adams et al., 2010*), with a search model based on coordinates from human AdoMetDC (PDB access code 3EP9) (*Bale et al., 2008*). Automated model building via the *AutoBuild* routine (*Terwilliger et al., 2008*) in *Phenix* yielded a model that contained 68% of all residues. Alternating cycles of manual model building in *Coot* (*Emsley et al., 2010*) were followed by standard positional and anisotropic atomic displacement parameter (ADP) refinement in *Phenix* (*Afonine et al., 2012*). Residues with missing or poor electron density (139–142, 159–161, 180–188, and 357–370) were not built into the structure. Pyruvoyl was added to the model after the initial refinement and included in further refinement with geometry restraints generated in *eLBOW* (*Moriarty et al., 2009*).

Initial phases for the TbAdoMetDC/prozyme complex were generated by molecular replacement. Briefly, two copies of the modified TbAdoMetDCΔ26 structure were aligned to the structure of the

human S68A processing mutant dimer (PDB access code 1MSV) (*Tolbert et al., 2003*) to create a putative heterodimer assembly, which was then used as a search model in *Phaser*, followed by initial model building with *AutoBuild*. The model was further improved by iterative cycles of manual model rebuilding in *Coot* and standard positional and TLS ADP refinement *in Phenix*. Electron density was missing or of poor quality for *Tb*AdoMetDC residues 1–4 (1-5), 24–26, 358–370 (357-370) and pro-zyme' residues 1'−2' (1'−4'), 25'−32' (25'−30'), 208'−218' (208'−218'), 239'−242' (239'−241'), 287'−290' (286'−290'), 325' (missing residues for the second molecule in the asymmetric unit are in parentheses). Pyruvoyl, Bis-tris propane and putrescine were incorporated into the model as described above.

The structure of *Tb*AdoMetDC/prozyme heterodimer with bound CGP 40215 was solved by molecular replacement in *Phaser* using the refined *Tb*AdoMetDC/prozyme complex structure as the search model. The model was further improved by iterative cycles of manual model rebuilding in *Coot* and standard positional and TLS ADP refinement in *Phenix*. Electron density was missing or of poor quality for *Tb*AdoMetDC residues 1–4 (1–4), 23–26, 357–370 (357-370) and prozyme' residues 1'−3' (1'−3'), 25'−31' (25'−31'), 207'−218' (207'−218'), 239'−242' (239'−241'), 286'−293' (286'−293'), 325'. Pvl, Bis-tris propane, putrescine and CGP 40215 were added to the model as described above.

Refined structures were analyzed in *MolProbity* (*Chen et al., 2010*). Atomic representations were created using *PyMOL Molecular Graphics System* (Version 1.7, Schrödinger). Secondary structure in cartoon representations was assigned with *DSSP* (*Kabsch and Sander, 1983*) and visualized using the *DSSP* plugin for *PyMOL* (by Hongbo Zhu, 2011, BIOTEC, TU Dresden). Structures were aligned using *TM-align* and RMSD was calculated as described (*Zhang and Skolnick, 2005*). Buried surface areas were calculated using the *PDBePISA* web server (*Krissinel and Henrick, 2007*).

### Sequence analysis

*T. brucei* prozyme protein sequence (XP_845564.1) was used to query the RefSeq_protein database with PSI-BLAST (*Cameron et al., 2004*) (default settings, 1000 maximum hits, 3 iterations) to identify AdoMetDC representatives (947 sequences). Identified sequences were submitted to batch CD-search (*Marchler-Bauer et al., 2015*) against the PFAM database to confirm the presence of an Ado-MetDC domain (pfam01536) and were analyzed according to taxonomic groups using batch Entrez on the NCBI server. AdoMetDC sequences were distributed in animals (348), plants (302), fungi (164), protists (65), and bacteria (68). The eukaryotic AdoMetDC sequences were submitted to the MAFFT server for multiple sequence alignment (*Katoh and Standley, 2013*).

## Acknowledgements

This work was supported by National Institutes of Health grants 2R37AI034432 and R01AI090599 (to MAP). MAP also acknowledges the support of the Welch Foundation grant I-1257. MAP holds the Beatrice and Miguel Elias Distinguished Chair in Biomedical Science and the Carolyn R Bacon Professorship in Medical Science and Education. Results shown in this report are derived from work performed at Argonne National Laboratory, Structural Biology Center at the Advanced Photon Source. Argonne is operated by UChicago Argonne, LLC, for the U.S. Department of Energy, Office of Biological and Environmental Research under contract DE-AC02-06CH11357. The authors thank Drs. Rama Ranganathan, Tony Michael and Yuh Min Chook for critical reading of the manuscript.

## Additional information

### Funding

| Funder | Grant reference number | Author |
|---|---|---|
| National Institute of Allergy and Infectious Diseases | 2R37AI034432 | Margaret A Phillips |
| National Institute of Allergy and Infectious Diseases | R01AI090599 | Margaret A Phillips |
| Welch Foundation | I-1257 | Margaret A Phillips |

The funders had no role in study design, data collection and interpretation, or the decision to submit the work for publication.

## Author contributions

OAV, Conception and design, Acquisition of data, Analysis and interpretation of data, Drafting or revising the article; LK, NG, MAP, Conception and design, Analysis and interpretation of data, Drafting or revising the article; CA, XD, SZ, DRT, ZC, Acquisition of data, Analysis and interpretation of data, Drafting or revising the article

## Author ORCIDs

Diana R Tomchick, http://orcid.org/0000-0002-7529-4643
Margaret A Phillips, http://orcid.org/0000-0001-5250-5578

## Additional files

### Supplementary files

• Supplementary file 1. Primers used in molecular cloning of the expression constructs.

• Supplementary file 2. Multiple sequence alignment of trypanosomatid prozymes with representative eukaryotic AdoMetDCs. Residue positions are highlighted according to conservation: mainly hydrophobic (yellow), mainly small (gray), invariant residues in AdoMetDC enzymes that likely contribute to catalysis (black), coevolving residues that likely contribute to activation (magenta), conserved trypanosomatid prozyme/enzyme residues that stabilize the prozyme/enzyme interaction (orange), and conserved trypanosomatid prozyme/enzyme residues that form the N-helix pocket (wheat). Sequences are labeled according to NCBI GenInfo Identifier (GI) followed by the species. Alignment sections are labeled above according to taxonomy group. Non-conserved termini and insertions are removed from some sequences, with the numbers of omitted insertion residues that belong to unconserved positions indicated with brackets. Aligned sequences were ordered according to taxonomy, with the exception of protist sequences, which were split according to their similarity to other sequences in the alignment (Sphaeroforma arctica and Salpingoeca rosetta sequences are closer to metazoa than other protists) and to distinguish the Trypanosomatid group that contains prozyme sequences. Trypanosomatids were supplemented with sequences from additional species present in the NR database (Phytomonas sp. isolate EM1, Angomonas deanei, and Strigomonas culicis), and indicated prozyme sequences were extended to the N- or C-termini using TBLASTN (marked by *). The confirmed AdoMetDC domains were found to be fused to several additional domains in select sequences, including an AdoMetDC leader (pfam08132) in 35 plant sequences, an f-box-like domain (pfam12937) in one fungal sequence, pyridoxal-dependent ornithine decarboxylase (pfam02784 and pfam00278) in 9 apicomplexan and 3 plant sequences, and protein prenyltransferase alpha subunit repeat (pfam01239) in one Blastocystis sequence. The proline residue (T. brucei P31) that alters peptide bond isomerization upon activation belongs to the Trypanosomatid AdoMetDC sequence motif "FEGPEK" and is present in all complete fungal sequences, with a single exception from Vanderwaltozyma polyspora (XP_001646102.1). P31 was also present in all but 19 protist sequences, where it was replaced in all apicomplexans (11), choanoflagellates (2), Blastocystis (2), Perkinsida (1), Ichthyophonida (1), Apusomonadidae (1), and Capsaspora (1). Inspection of the multiple sequence alignment identified several residues that appeared to coevolve with the P31 in protist sequences (that have not lost the proline), including T104, N132, H172, and C269. The H172 coevolution also extends throughout fungal sequences, while the T104, N132, and C269 are restricted to a more limited subset. These residues occupy key positions in TbAdoMetDC with respect to the conformational changes that occur upon prozyme activation as described in the main manuscript.

### Major datasets

The following datasets were generated:

| Author(s) | Year | Dataset title | Dataset URL | Database, license, and accessibility information |
|---|---|---|---|---|
| Volkov OA, Ariagno C, Chen Z, Tomchick DR, Phillips MA | 2016 | Crystal structure of Trypanosoma brucei AdoMetDC-delta26 monomer | http://www.rcsb.org/pdb/explore/explore.do?structureId=5TVO | Publicly avaliable at the RCSB Protein Data Bank (accession no. 5TVO) |
| Volkov OA, Chen Z, Tomchick DR, Phillips MA | 2016 | Crystal structure of Trypanosoma brucei AdoMetDC/prozyme heterodimer | http://www.rcsb.org/pdb/explore/explore.do?structureId=5TVM | Publicly avaliable at the RCSB Protein Data Bank (accession no. 5TVM) |
| Volkov OA, Chen Z, Tomchick DR, Phillips MA | 2016 | Crystal structure of Trypanosoma brucei AdoMetDC/prozyme heterodimer in complex with inhibitor CGP 40215 | http://www.rcsb.org/pdb/explore/explore.do?structureId=5TVF | Publicly avaliable at the RCSB Protein Data Bank (accession no. 5TVF) |

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
