## [Decision Letter]

Thank you for submitting your article "Relief of autoinhibition by conformational switch explains enzyme activation by a catalytically dead paralog" for consideration by *eLife*. Your article has been reviewed by three peer reviewers, and the evaluation has been overseen by John Kuriyan as the Senior and Reviewing Editor. The following individual involved in review of your submission has agreed to reveal her identity: Catherine L Drennan (Reviewer #3).

The reviewers have discussed the reviews with one another and the Reviewing Editor has drafted this decision to help you prepare a revised submission.

Phillips and co-workers present a series of crystal structures that reveal how an inactive enzyme paralog can activate its partner enzyme through heterodimer formation. The structural changes that the authors have discovered are substantial and fascinating. Specifically, they previously identified an inactive paralog of S-adenosylmethionine decarboxylase (AdoMetDC) in the genome of Trypanosoma brucei. They also reported that the activity of TbAdoMetDC is greatly increased when mixed with the inactive paralog, which they call prozyme. The current work investigates the structural basis for the increase in activity. Three X-ray structures were reported: 1) an N-terminal truncation of the TbAdoMetDC monomer, 2) the TbAdoMetDC/prozyme heterodimer, and 3) the heterodimer with bound to CGP, a known inhibitor of AdoMetDC. The authors further explore the role of putrescine, a known activator of human ADoMetDC.

In the structure of the inactive monomer, residues 27-30 block the active site, suggesting an autoinhibitory role of the sequence. A novel putrescine binding site was identified near the N-terminal helix of the active TbAdoMetDC monomer in the heterodimer structures and report mutagenesis and kinetic studies that are consistent with a regulatory role of the putrescine. The binding site for putrescine showed significant differences compared to that of HsAdoMetDC in terms of both location and residue composition. Bis-tris propane was also observed to bind to the prozyme subunit.

The authors conclude that the N-terminus of TbAdoMetDC is involved in autoinhibition and that prozyme induces conformational changes during heterodimer formation that relieves this inhibition, resulting in an active form of the enzyme. They further explore the role of putrescine in the activation of TbAdoMetDC.

There is considerable interest in "pseudoenzymes" and of the existence of heterodimers in which one chain is inactive. This work provides an interesting explanation for why such paralogs may have been retained. It is also interesting that this enzyme family may have evolved such that small molecules could perform the regulatory role of the paralog using a similar molecular mechanism as the paralog. In this respect, the putrescine binding site right near helix h1 is interesting.

The medical relevance of this work is also potentially exciting. These studies show differences between human S-adenosylmethionine decarboxylases (AdoMetDCs) and AdoMetDCs from Neglected Topical Disease causing organisms (i.e. organisms that cause sleeping sickness, Chagas disease and Leishmaniasis). With this structural data, one can pursue the design of specific inhibitors.

Please address the following points in a revised manuscript. No additional experimentation is called for.

1) The authors determined the structure of an N-terminal truncation of TbAdoMetDC and conclude that because residues 27-30 block the active site, that this would also happen in the full length TbAdoMetDC. What evidence supports this conclusion?

2) As noted above, the putrescine binding site is of interest. The authors observe binding of bis-tris propane at the putrescine binding site of TbAdoMetDC. Aside from the diamine core, bis-tris propane is not very similar to putrescine and the relevance of its binding at 100 mM concentration is not clear. Why does it bind to the TbAdoMetDC site but not the corresponding prozyme site, which instead binds putrescine? What is the binding constant for bis-tris propane? Does bis-tris propane stimulate TbAdoMetDC activity? Does bis-tris propane bind to the inactive TbAdoMetDC monomer? In short, how do the authors distinguish between a crystallization artifact and relevant binding?

3) There is some disagreement among the reviewers as to the quality of communication in the manuscript. One of the reviewers has this to say:

"The structural changes that the authors have discovered are substantial, fascinating, and beautifully described. The writing is excellent and so are the figures. It was a joy to read this manuscript, which provides an excellent example of how crystallographic work should be presented."

However, the other two reviewers and the editor agree that the figures and the text are could be improved so as make it easy for non-specialists to follow. The figures, in particular, tend to lose the reader, and figure legends are sometimes confusing. Color coding is complex and so need to be clearly defined for the multiple panels of a figure. The manuscript would be of more general interest if the interesting structural concepts would be more clearly communicated. The presentation of the structural data is challenging since the proteins are complex and some of the key points of the paper (the three step activation mechanism) could be more clearly communicated. While the paper is written in a classic crystallographic style, the manuscript would be improved greatly by the addition of simplified schematic diagrams. The arguments about the evolution of the allosteric mechanism are simple enough to follow from the text, but the illustrations do not make it easy for the reader to map the relevant ideas onto the structure.

In summary, this paper will fail to reach its potential readership if the interesting structural concepts are not more clearly communicated.

---

## [Author Response]

[…]

*Please address the following points in a revised manuscript. No additional experimentation is called for.*

*1) The authors determined the structure of an N-terminal truncation of TbAdoMetDC and conclude that because residues 27-30 block the active site, that this would also happen in the full length TbAdoMetDC. What evidence supports this conclusion?*

The structure of the *Tb*AdoMetDC truncated monomer shows that the autoinhibitory sequence makes a number of good contacts in the active site, including H-bonds and π stacking interactions that are consistent with good binding affinity and thus effective autoinhibition. In addition to the structural data showing that these residues are bound in the active site of the N-terminal truncated *Tb*AdoMetDC, the biochemical and structural data on the full-length heterodimer also support the conclusion that residues 27-30 are autoinhibitory. First our mutagenesis data show that truncation or specific point mutation of the N-terminal 16 amino acids of AdoMetDC lead to loss of prozyme induced activation. Secondly this N-terminus is highly conserved in the trypanosomatids. These data implicate the N-terminus in the activation mechanism. The X-ray structural data of the full-length heterodimer shows that the N-terminus is bound into the interface, and its position in the interface effectively stabilizes the open conformation of the active site where the autoinhibitory sequence has been displaced. The displacement of the autoinhibitory sequence out of the active site in the full-length heterodimer (relative to the truncated monomer) is only one of the structural changes that were observed. Docking of the N-terminus is accompanied by major structural rearrangements that are required to form the N-terminal helix binding site and which also promote dimer contacts across the interface. These include β-strand reorganization and disordered to ordered loop transitions. Thus, the observed conformational changes are part of a coupled process involving not only the autoinhibitory sequence, but also these other conformational changes, supporting our conclusion that the inactivity of the monomeric structure is based on autoinhibition by residues near the N-terminus. The existence of a proline residue that is conserved in the trypanosomatids, and which undergoes a *cis-trans* conformational change from the monomer to dimer, also supports our autoinhibition mechanism since *cis-trans* proline isomerization has been observed in other auto-inhibitory sequences in other proteins and it is conserved in all of the trypanosomatids that utilize this allosteric mechanism, again supporting its role as described. So in short the highly coupled nature of the multiple conformational changes that were observed, all of which are centered on building a binding site for the N-terminus and for generating new contacts across the dimer interface and the conservation of the residues that are involved, make it highly unlikely that the observed auto-inhibition of the truncated monomer would be an artifact. We have added a discussion of these points into the manuscript into the first paragraph of the Results section and into the first paragraph of the Discussion section.

*2) As noted above, the putrescine binding site is of interest. The authors observe binding of bis-tris propane at the putrescine binding site of TbAdoMetDC. Aside from the diamine core, bis-tris propane is not very similar to putrescine and the relevance of its binding at 100 mM concentration is not clear. Why does it bind to the TbAdoMetDC site but not the corresponding prozyme site, which instead binds putrescine? What is the binding constant for bis-tris propane? Does bis-tris propane stimulate TbAdoMetDC activity? Does bis-tris propane bind to the inactive TbAdoMetDC monomer? In short, how do the authors distinguish between a crystallization artifact and relevant binding?*

We had discussed in the original manuscript the observation that the putrescine binding site in AdoMetDC was larger than in prozyme. We think that this larger pocket likely has a lower affinity for putrescine than the pocket in prozyme and so because of the reduced affinity it was outcompeted by the buffer, which was present at 25-fold excess the concentration of putrescine. So indeed we do think that the binding of bis-tris was an artifact of the crystallization conditions and that the natural ligand is likely to be putrescine. This hypothesis is also supported by prior mutagenesis data on the *T. cruzi* enzyme. We have rewritten this section of the Results to provide clarification of these points.

*3) There is some disagreement among the reviewers as to the quality of communication in the manuscript. One of the reviewers has this to say:*

*"The structural changes that the authors have discovered are substantial, fascinating, and beautifully described. The writing is excellent and so are the figures. It was a joy to read this manuscript, which provides an excellent example of how crystallographic work should be presented."*

*However, the other two reviewers and the editor agree that the figures and the text are could be improved so as make it easy for nonspecialists to follow. The figures, in particular, tend to lose the reader, and figure legends are sometimes confusing. Color coding is complex and so need to be clearly defined for the multiple panels of a figure. The manuscript would be of more general interest if the interesting structural concepts would be more clearly communicated. The presentation of the structural data is challenging since the proteins are complex and some of the key points of the paper (the three step activation mechanism) could be more clearly communicated. While the paper is written in a classic crystallographic style, the manuscript would be improved greatly by the addition of simplified schematic diagrams. The arguments about the evolution of the allosteric mechanism are simple enough to follow from the text, but the illustrations do not make it easy for the reader to map the relevant ideas onto the structure.*

*In summary, this paper will fail to reach its potential readership if the interesting structural concepts are not more clearly communicated.*

Thank you for the recommendations and comments. We agree that the structural changes are substantial, fascinating and beautiful, but we take the point that they could have been better illustrated. We have made the following changes in the figures to hopefully more clearly illustrate the results.

1) We have changed the color of prozyme from gray to yellow. We realized that the gray was a bit too similar to the blue we were using for the AdoMetDC subunit, which made it difficult to distinguish between them in some of the comparative figures.

2) We have added diagrams above the key figures to show the color codes used for the various chains that are depicted in the figure.

3) We have added a new figure (Figure 8), which depicts a cartoon illustration of the structural changes that we observed upon prozyme binding.

4) We have recolored Figure 7, the evolutionary tree to better illustrate the steps that lead to formation of the eukaryotic enzymes.

5) We have added additional descriptions of the colors used in the figures into the figure legends.

We have added an additional paragraph to the start of the Results section that summarizes the structural findings, and added a few additional points into the Introduction to better set up the concepts that will follow in the rest of the manuscript.